# Polygenic adaptation of rosette growth in *Arabidopsis thaliana*

**Benedict Wieters[1], Kim A. Steige[1], Fei He[1], Evan M. Koch[2,3], Sebastián E. Ramos-Onsins[4], Hongya Gu[5], Ya-Long Guo[6], Shamil Sunyaev[2,3], Juliette de Meaux[1]\***

**1** Institute of Botany, University of Cologne, Cologne, Germany, **2** Genetics Division, Brigham & Women's Hospital and Harvard Medical School, Boston MA, United States of America, **3** Department of Biomedical Informatics, Harvard Medical School, Boston MA, United States of America, **4** Centre for Research in Agricultural Genomics (CRAG) CSIC/IRTA/UAB/UB, Bellaterra, Spain, **5** State Key Laboratory for Protein and Plant Gene Research, College of Life Sciences, Peking University, Beijing, China, **6** State Key Laboratory of Systematic and Evolutionary Botany, Institute of Botany, Chinese Academy of Sciences, Beijing, China

\* jdemeaux@uni-koeln.de

**Data Availability Statement:** All relevant data are within the manuscript and its Supporting Information files. Raw image data and image analysis scripts are stored in the DRYAD repository (doi:10.5061/dryad.s1rn8pk5m).

## Abstract

The rate at which plants grow is a major functional trait in plant ecology. However, little is known about its evolution in natural populations. Here, we investigate evolutionary and environmental factors shaping variation in the growth rate of *Arabidopsis thaliana*. We used plant diameter as a proxy to monitor plant growth over time in environments that mimicked latitudinal differences in the intensity of natural light radiation, across a set of 278 genotypes sampled within four broad regions, including an outgroup set of genotypes from China. A field experiment conducted under natural conditions confirmed the ecological relevance of the observed variation. All genotypes markedly expanded their rosette diameter when the light supply was decreased, demonstrating that environmental plasticity is a predominant source of variation to adapt plant size to prevailing light conditions. Yet, we detected significant levels of genetic variation both in growth rate and growth plasticity. Genome-wide association studies revealed that only 2 single nucleotide polymorphisms associate with genetic variation for growth above Bonferroni confidence levels. However, marginally associated variants were significantly enriched among genes with an annotated role in growth and stress reactions. Polygenic scores computed from marginally associated variants confirmed the polygenic basis of growth variation. For both light regimes, phenotypic divergence between the most distantly related population (China) and the various regions in Europe is smaller than the variation observed within Europe, indicating that the evolution of growth rate is likely to be constrained by stabilizing selection. We observed that Spanish genotypes, however, reach a significantly larger size than Northern European genotypes. Tests of adaptive divergence and analysis of the individual burden of deleterious mutations reveal that adaptive processes have played a more important role in shaping regional differences in rosette growth than maladaptive evolution.

**Funding:** This research was funded by the European Research Council (ERC) through the "AdaptoSCOPE" grant 648617 to JdM, by the German American Fulbright Commission to JdM, and by grant AGL2016-78709-R (MEC, Spain) to S. E.R.-O. S.E.R.-O also acknowledges the financial support of the Spanish Ministry of Economy and Competitivity for the Center of Excellence Severo Ochoa 2016–2019 (SEV-2015-0533) and by the CERCA Programme/Generalitat de Catalunya. We further acknowledge grants R35 GM127131 and RO1 MH101244 from the National Institutes of Health (NIH) for S.S and E.M.K. The funders had no role in study design, data collection and analysis, decision to publish, or preparation of the manuscript.

**Competing interests:** The authors have declared that no competing interests exist.

## Author summary

The rate at which plants grow is a major functional trait in plant ecology. However, little is known about its genetic variation in natural populations. Here, we investigate genetic and environmental factors shaping variation in the growth rate of *Arabidopsis thaliana* and ask whether genetic variation in plant growth contributes to adaptation to local environmental conditions. We grew plants under two light regimes that mimic latitudinal differences in the intensity of natural light radiation, and measured plant diameter as it grew over time. When the light supply was decreased, plant diameter grew more slowly but reached a markedly larger final size, confirming that plants can adjust their growth to prevailing light conditions. Yet, we also detected significant levels of genetic variation both in growth rate and in how the growth dynamics is adjusted to the light conditions. We show that this variation is encoded by many loci of small effect that are hard to locate in the genome but overall significantly enriched among genes associated with growth and stress reactions. We further observe that Spanish genotypes tended to reach, on average, a significantly larger rosette size than Northern European genotypes. Tests of adaptive divergence indicate that these differences may reflect adaptation to local environmental conditions.

## Introduction

Growth rate is a crucial component of individual fitness, as it reflects the capacity of the organism to acquire resources and conditions reproductive output [1,2]. In experimental evolutionary studies, relative growth rate provides a measure of microbial adaptation in response to selection [3]. In plants, however, little is known about the evolutionary processes that influence variation in plant growth rate, despite its cornerstone importance in plant ecology [4–6].

Four processes may explain variation in growth rate: random evolution due to drift, plasticity, adaptation or maladaptation. Plasticity describes the immediate adjustment of plant growth rate in response to environmental modifications [7]. Such change may occur as a passive consequence of resource limitations. Plant growth, for example, is slower in drought conditions or at lower temperatures [8,9]. Plastic adjustments of plant growth, however, can also actively contribute to maintaining fitness under challenging conditions. For example, shade avoidance allows plants to outgrow neighbors competing for light [10]. Such reactions may allow the organism to maintain high fitness when the environment becomes challenging, without having to evolve genetically [11].

As the distribution range of a species expands, plastic modifications may become insufficient to adjust fitness, and genetic variation may be required for local adaptation [7,12]. There is clear evidence that genetic variation in plastic life history traits such as flowering time or seed dormancy contributes to the evolution of life-history decisions that are tailored to the local optimal growth season [13–16]. Surprisingly, the extent to which genetic variation in plant growth rate itself contributes to local adaptation is not known. Answering this question requires that the effect of natural selection on phenotypic divergence be disentangled from the effect of drift [17].

Genetic variation in growth rate may also arise in the absence of a compelling environmental change, as a consequence of population genetics processes. In bottlenecked populations, or in the aftermath of rapid range expansion, increased drift hampers the efficient removal of deleterious mutations, and individuals may become less fit [18–22]. Because plant growth is a component of fitness, genotypes carrying a larger burden of deleterious mutations may show

decreased growth. Genetic variation in growth rates may thus also reflect maladaptation resulting from decreased population size.

The annual species *A. thaliana* has become a model system for both molecular and evolutionary biology, and it is well suited for determining the ecological and evolutionary significance of plant growth rates [23,24]. *A. thaliana* individuals can adjust their growth rate plastically to maintain their fitness. Plant rosettes grow to a larger diameter when light becomes limited [10,25]. Ample genetic variation in plant growth rates has also been documented in this species [26–28]. In addition, there is evidence that the resources allocated to growth are not identical throughout the species' range, because trade-offs between growth rate and development change with latitude (reviewed in [12,29,30]). Furthermore, traits related to how resources are allocated to growth, such as growth inhibition upon the activation of plant defense, or plant dwarfism, have been associated with adaptation [31–35]. In summary, adaptive variation in the rate of plant growth may have evolved in *A. thaliana*. At the same time, the maladaptive or neutral evolution of a decreased growth rate cannot be excluded *a priori*. Indeed, *A. thaliana* has experienced recent severe bottlenecks in parts of its range, such as in Northern European or Chinese populations, which locally increased the rate of genetic drift and led to an accumulation of deleterious genetic variants [36–38]. Neutral evolutionary forces could therefore also have modified growth rate in these populations.

To determine the roles of deleterious variation, adaptive evolution and/or plasticity in the genetic variation among plant growth rates, we analyzed variation among rosette growth rates across genotypes sampled from four broad regions (China, Spain, Northern and Western Europe). To assess the relative roles of genetic and plastic variation, we grew plants under two light regimes that mimicked constitutive latitudinal differences in natural light intensity and characterized genetic variation in growth plasticity. This analysis reveals significant regional differences in growth dynamics, most of which have a polygenic basis. Population genetics analyses indicate that local selective pressures have helped shape this variation.

## Materials and methods

### Phenotypic analysis and estimation of growth rate parameters

We chose 278 genotypes of *Arabidopsis thaliana* originating from 220 locations distributed throughout 4 regions for phenotypic analyses of growth rate variation (Northern Europe, Western Europe, Spain and Central-Eastern China, S1 Table and S1 Fig). A PCA confirmed that genotypes within these regions formed distinct phylogeographic clusters (S2 Fig), whose specific evolutionary history has been previously documented [13,37–39].

Seeds were stratified for 3 days at 4˚C in the dark on wet paper, and six replicate seedlings per genotype were replanted, each in one 6x6 cm round pots containing soil ("Classic" from Einheitserde) mixed with perlite. Growth was measured in a split-plot design, under two light regimes, high light (HL) and low light (LL) in the same chamber but in successive independent trials. Plants were grown in a temperature-controlled walk-in growth chamber (Dixell, Germany) set at 20˚C day and 18˚C night, and watered once a week. For each light regime, pots were randomized within three blocks of 8 trays with 7x5 pots, with one replicate of each genotype in each block. Trays were randomized and the rows in the trays were rotated every two to three days to account for variability within the chamber. The plants were exposed to light for 12 h with LEDs (LED Modul III DR-B-W-FR lights by dhlicht) set to 100% intensity of blue (440nm), red (660nm) and white (HL conditions) or 30% of red and blue plus 100% of white light (LL conditions), followed by a 10 min far-red light pulse to simulate sunset (40% intensity at 735nm). The total measured light intensity was 224 +/- 10 $\mu mol/m^2 s$ in HL and 95 +/- 7 in LL. These two light regime mimick latitudinal differences in natural light intensity (S3 Fig).

Individual plants were photographed approximately bi-weekly with a Canon EOS 5D Mark III digital camera until days 46 (8 weeks) and 89 (13 weeks), for those grown under the HL and LL regimes, respectively (image data is available on dryad, doi:10.5061/dryad.s1rn8pk5m) [40]. We only measured diameter for one time point per week, but included additional measurements if it was necessary to fit the logistic curves. Flowering time was measured as days to first flower opening. For genotypes without a flowering individual by the end of the experiment, a flowering time value of 59 or 90 days (last date that flowering was scored) was assigned to HL and LL plants, respectively. Since only 37% of the plants in the experiment flowered, we also used flowering time data from the 1001 Genomes project, according to which flowering was scored at 10 and 16˚C for 177 genotypes [39]. A measure of the diameter of each plant (defined as the longest distance between two leaves) was extracted at least once a week with ImageJ (v.1.50b, [41]). In a preliminary experiment conducted on a subset of 17 genotypes, we used *Rosette Tracker*, an ImageJ tool [42], to show that diameter correlated positively with rosette area under both light regimes (HL r = 0.83, p<3.2e-5, LL r = 0.56, p<0.0186). We confirmed that plant diameter accurately predicts rosette area on this larger data set (r = 0.929, p = <2.2e-16 in HL). Rosette diameter was therefore used to determine the increase in rosette area over time.

We conducted two additional experiments to test the ecological relevance of rosette growth variation measured under controlled conditions. First, all genotypes were grown under HL conditions, in 5 replicates. In this experiment, instead of rosette diameter, we measured hypocotyl length after 15 days, to quantify variation in seedling growth. We further weighted 3-week old plants with a precision balance (Sartorius AC 210 P with accuracy of 0.1 mg) to quantify variation in plant biomass. We also followed a similar experimental design to measure the diameter of plants grown outdoor in 2 replicates in the field of the Cologne Institute of Plant Sciences. Sand was used instead of soil in 9 cm diameter pots. Seeds were sown in September 2016, which corresponds to the native season in the area and put outside after a week.

## Statistical analysis of genetic variance

All following analyses were conducted using R (version 3.6.3) [43], and function names refer to those in the R package mentioned unless otherwise noted. We provide an Rmarkdown script detailing the statistical analysis of phenotypic variation (S1 File).

The split-plot design of our study allowed us to conduct the analysis in successive steps. First, we extracted three parameters that together provided a comprehensive description of individual rosette growth. For this, rosette diameter measurements over time (our input phenotype) were modeled as a three-parameter logistic growth using the *drm* function from the *drc* package in R [44]. The three following growth parameters were extracted: final size (FS, largest estimated rosette diameter in cm), slope (factor of magnification in the linear phase) and t50 (inflection point; time at which growth is maximum and half of FS has been reached, which quantifies the duration of the rosette area growth phase in number of days). We show examples for the estimation of growth rate in S4 Fig For each parameter, a genotypic mean correcting for block, tray and position effect was computed with a generalized linear model with a Gaussian error distribution and the following model: *parameter~accession+block+tray/(row+col)+error*. Genotypic means in HL and LL were extracted separately, because light treatments had to be performed in separate trials. To quantify the plasticity of growth to the light regime of each genotype, we correlated genotypic means in LL against HL and extracted the residuals. GxE estimates thus quantify the deviation of the response of a given genotype from the mean response of all genotypes for the respective parameter. The estimate increases with the magnitude of growth plasticity induced by a decrease in light intensity (S5 Fig). Broad-

sense heritability ($H^2$) was determined for each trait in each environment as previously reported [45]. Briefly, genetic and environmental variances were estimated using the *lme* function from the *nlme* package [46], with the block as fixed and genotype as random effect and heritability was determined as the ratio of genetic variance over the total variance. The heritability of GxE was not estimated, because we quantified plasticity on the basis of changes in genotypic means in the two light conditions. For this reason, we also computed trait pseudo-heritability, which is based on the genome-wide association study (GWAS) mixed model (see below) and allowed us to estimate the proportion of the observed phenotypic variance that is explained by genotypic relatedness for all traits [47].

To assess the correlation of phenotypic traits with climatic variables, we investigated solar radiation estimates, temperature, precipitation, humidity and wind speed with 2.5-min grid resolution (WorldClim2 database, [48], accessed on March 20, 2018) and soil water content [49]. Following [45], we estimated the mean over the putative growth season for each genotype in addition to the annual averages.

Because of the strong correlations between climatic variables, we conducted principal component analyses (PCAs) to combine the data. We analyzed annual average radiation separately and combined the other variables into the PCAs: growing season data, variables related to precipitation and to temperature. Raw climatic data and the principle components (PCs) are in S1 Table and the loadings of the PCA are in S11 Table.

Regional differences in mean growth parameters were tested with a multivariate analysis, using the *manova* function, the matrix of growth parameters (genetic means) or plasticity and the following model: *growth~ population\*light regime*. Significance levels were determined by the Pillai test.

For univariate analysis, we used GLMs to test the effect of population of origin on the genotypic means. A Gaussian distribution was taken for error distribution, and the dispersion parameter was estimated by the *glm* function. Group means were compared with the *glht* function (which performs general linear hypothesis testing) and plotted on boxplots using the *cld* function, both of the *multcomp* package [50].

Pairwise trait correlations within and across populations were calculated with the *cor.test* function (Pearson's product-moment correlation), and p-values were established using the *lmekin* function in R, which includes a kinship matrix of individuals (see below) and thus corrects for population structure (after [29]). We used the *corrplot* function from the *corrplot* package to plot correlations [51]. Plots were modified using inkscape version 0.92.3 (inkscape. org, [52]). Significance levels were adjusted for false-discovery rates with the function *p.adjust*.

## Genome-wide association studies

Genomic data were available for 231 of the 278 genotypes included in the phenotypic analysis, i.e. for 84 genotypes from Northern Europe (NE, predominantly Sweden), 3 from Western Europe (WE), 119 from Spain (SP) and 15 from China (CH) [38,39]. Chinese genotypes were excluded from the GWAS because of their limited number (15 genotypes) and their strong genetic divergence ([38], S2 Fig). In total, the growth parameters of 203 and 201 genotypes grown under HL and LL, respectively, were used for the GWAS. Genome-wide association studies were conducted using the method from [53]. The corresponding GWAS package was downloaded from *https:// github.com/arthurkorte/GWAS*. Single Nucleotide Polymorphisms (SNPs) with minor allele frequency below 5% or with more than 5% missing data were removed from the genotype matrix, resulting in a matrix of 1,448,192 SNPs, produced with *vcftools* (—012 recode option) (version 0.1.15, [54]). A kinship matrix was computed with the *emma.kinship* function. For each growth parameter, genotypic means were used as phenotype measurements. We performed GWAS

across the European sample of genotypes but also within region (Spain and Northern Europe). For each SNP, the script output delivered p-values, which were Bonferroni corrected for multiple testing, and effect sizes. For each trait, we estimated pseudo-heritability, the proportion of the observed phenotypic variance that is explained by the estimated relatedness (e.g. kinship matrix, [47]). To identify candidate genes underpinning significant GWAS associations, we calculated the linkage disequilibrium (LD) around the SNP of interest and selected all genes that were in a genomic window with LD above 0.5 within a 250 kb window around the SNP. The LD was calculated as the Pearson correlation between the frequencies of allele pairs. Additionally, we downloaded an annotation of loss-of-function (LOF) variants [55] and performed a GWAS association following the procedure described above except that the SNP data set was replaced with the LOF data set, which assigned one of two states (functional or LOF), for each genotype and each of the 2500 genes with known LOF alleles.

## Validation of the polygenic signal

In humans, where population structure and environmental variation are correlated, insufficient correction of the genetic associations caused by shared ancestry has been shown to create spurious associations [56–58]. Even though environmental variance is much better controlled in common garden experiments including kinship as a covariate, association tests can still be confounded by genetic relatedness [57]. This is of particular concern when many trait/SNP associations are below the Bonferroni significance threshold. The rate of false positive was not excessively inflated by genomic differentiation between regions, because GWAS performed within regions (Northern Europe or Spain) had similar p-value distributions than GWAS performed on the complete phenotypic dataset (S6 and S7 Figs). We nevertheless used two additional approaches to confirm the polygenic basis of traits. First, we examined whether phenotypic variation could be predicted by polygenic scores derived from sub-significant GWAS hits, with $p<10^{-4}$. To this end and for each trait, we calculated genotypic means, performed GWAS, as described above, and computed polygenic scores following [53]. SNPs with a significant association with the phenotype were pruned to remove SNPs standing in strong linkage disequilibrium with *plink* version 1.90 [59], following [56]. The *plink -clump* function was set to select SNPs below a (GWAS) P- value threshold of 0.0001, start clumps around these index SNPs in windows of 1 Mb, and remove all SNPs with P < 0.01 that are in LD with the index SNPs. The SNP with the lowest p-value in a clump was retained for further analysis. Briefly, input files, including allele frequencies for all SNPs, all SNPs with GWAS p-values lower than $10^{-4}$ and their effect size estimates were created with a custom R script. We defined each genotype as its own population (genotypes from Spain and Northern Europe were grouped in regions). Scripts were downloaded from *https://github.com/jjberg2/ PolygenicAdaptationCode*. The pipeline was run with default parameters, and polygenic scores (Z-scores) were estimated. We used three approaches to validate the relevance of GWAS associations for predicting the phenotype. First, we used 80% of the genotypes and used the resulting GWAS association to compute a polygenic score for the remaining genotypes. Second, we took two replicates to compute the polygenic scores, and tested whether it predicted the phenotype of the third replicate. Third, we correlated the phenotypic values predicted by polygenic scores (calculated this time on the basis of all three replicates) with the observed phenotypic value. We repeated this replacing the set of SNP associated at $p<10^{-4}$ with 1000 sets of an equal number of randomly chosen SNPs. We then compared the correlation of polygenic scores to the input phenotype for SNPs associated at subsignificant level ($p<10{-4}$) to the correlation expected for random sets of SNPs. Correlations were calculated with the R function *cor.test* (Pearson's product-moment correlation).

In addition, we investigated functional enrichment among genes within 10kb of GWAS associated SNPs. To assign a single GWAS p-value for each gene, either we assigned for each trait the lowest p-value of SNPs within the gene, or, if no SNP was within the gene, we assigned the p-value from the physically closest SNP [47]. When there were GWAS hits in the vicinity of duplicated genes, we removed tandem duplicated genes within a 10-gene sliding window. For this, we first aligned all TAIR10 genes against each other by using BLAST (version 2.9.0, available at *https://blast.ncbi.nlm.nih.gov*). Then the duplicated genes were selected as genes with an e-value <1e-30. Finally, tandem duplicated genes identified with gene distance <10-genes were filtered out to avoid inflated functional enrichments. If the polygenic signal only due to insufficient correction for population structure, we expect that similar functions will be enriched among population structure outliers and among genes with low GWAS p-values. We thus computed Fst-values for each gene with the *F_ST.stats* function of the PopGenome library [60]tween Spain and Northern Europe. Negative Fst values were set to zero.

Enrichments were tested as previously described [61]. To call GO enrichment significant, we determined the conservative threshold $p = 0.008$. This threshold was determined as the 0.01% quantile of the p-value distribution when GO enrichments were tested for 1000 random sets of the same number of SNP. To assess similarity between traits in Gene Ontology (GO) enrichments, we calculated graph-based similarity with the *GOSemSim* package [62]. A distance matrix was estimated with average connectivity between the GO terms. The clustered GO categories were then plotted as a dendrogram with the *plot.phylo* function from the *ape* package (version 5.3, [63]). GO categories enriched at p-value below 0.001 were highlighted. The distribution of enriched GO categories was evaluated by visual inspection.

## Testing for adaptation or maladaptation

For population genetics analyses, we sampled one genotype at random whenever plants were sampled in the same location, acquiring a total of 220 genotypes. As a proxy for the genomic load imposed by deleterious mutations, the number of derived non-synonymous mutations per haploid genome has been proposed [64]. This approach was not possible here because the genomes of individuals from China and Europe were sequenced in different labs, and the depth and quality of sequencing varied too much to make a fair comparison. Instead, we used two data sets that together catalogued LOF alleles after controlling as much as possible for heterogeneity in sequencing quality: one that included European genotypes [55] and a more recent data set that included Chinese genotypes [65]. As an estimate of the individual burden of deleterious mutations, we counted the number of LOF alleles for each individual and tested whether individuals with a larger number tended to have a lower growth rate using the Spearman rank correlation.

To search for footprints of adaptive evolution, we computed an Fst value between Spain and Northern Europe for each SNP in the GWAS analysis using the R-package *hierfstat* and the *basic.stats* function [66]. Negative Fst values were set to zero, and the *quantile* function was used to calculate the 95[th] percentile. The Fst distribution of SNPs associated with any GWAS (p<10e-4) was compared to the genome-wide distribution with a Kolmogorov-Smirnov test. We also computed the likelihood that its 95th percentile was greater than the 95th percentile of 10 000 random samples of an equally large set of SNPs. To compare the phenotypic differentiation of traits, Qst values for the phenotypic traits were estimated as previously described [45]. Briefly, Qst was estimated as VarB / (VarW + VarB), where VarW is the genotypic variance within and VarB between regions. These variances were estimated with the *lme* function of the *nlme* package [46], with the block as fixed and population/genotype as random effect. We extracted the intercept variance for VarB and the residual variance for VarW. Since replicates

were taken from the selfed progeny of each genotype, VarB and VarW are broad-sense genetic variance components. To reveal signatures of local adaptation, the Qst of each trait was compared to the 95[th] percentile of the Fst distribution (between Spain and Northern Europe) [67,68]. We verified that outlier Qst values were unlikely to arise randomly. For this, we permuted phenotypic data by randomizing genotype labels and verified that the difference between observed Qst and 95[th] percentile of Fst was significantly greater than for randomized Qst, following [45]. In a second approach, we used a multivariate normal distribution to generate phenotypic divergence based on the kinship matrix to generate an expected Qst distribution [69]. Finally, we applied the over-dispersion test (Qx test), which compares polygenic scores computed for associated versus random SNPs (null model), in a process similar to a Qst/Fst comparison, but assuming that each population is composed of the selfing progeny of one genotype [53]. A Qx significantly larger than the Qx computed for the null model indicates that polygenic trait prediction is more differentiated than expected from the kinship matrix and can be taken as an indication that the trait has evolved under divergent selection, either within or between regions [53].

## Results & discussion

### Ecological relevance of rosette growth variation

On the basis of the more than 15,000 rosette images we collected, we used rosette diameter as a proxy to describe rosette growth variation with three parameters; each refers to the ways in which growth can differ among genotypes: i) the time until the exponential growth phase is reached (t50), ii) the speed of growth during the linear growth phase (slope) and iii) the final size (FS) at which rosette diameter plateaus at the end of the rosette growth phase (Fig 1 and S2 and S3 Tables). Of the parameters, FS displayed the highest broad-sense heritability, in plants grown under both regimes: high light (HL, $H^2$ = 0.636) and low light (LL, $H^2$ = 0.794, S4 Table). Trait variation measured in controlled settings sometimes fails to reflect variation expressed in natural conditions [70,71]. This is not the case for rosette growth variation in *A. thaliana*. FS in HL conditions correlated positively with plant biomass (r = 0.267, p-value = 4.6e-5) and seedling growth (r = 0.372, p-value = 9.1e-7) in the growth chamber. FS measured in HL also correlated positively with plant diameter measured under natural light in the field (r = 0.263, p-value = 0.0009). This indicates that a significant part of the variation we report is ecologically relevant.

### Environmental plasticity has the strongest impact on plant growth variation

Light regimes revealed that plasticity has the strongest impact on rosette growth (Fig 1, MANOVA HL vs LL: F = 2275.37, df = 1, p-value = <2.2e-16, Table 1). In plants grown under LL, the maximum growth rate was delayed and rosette growth plateaued at a larger size (Table 1 and Fig 2). This observation was in agreement with the reduced relative growth rate reported in many plant species when light supply decreases, whereas the larger FS reflected the expected shade avoidance reaction [72]. We observed that plants reached a larger diameter (and rosette area) by elongating their petiole and minimizing leaf blade overlap in LL, a reaction known as the shade avoidance response. This strong modification of leaf shape may explain the predominant impact of environmental variation we report here (Table 1). Nevertheless, we detect significant levels of genetic variation in growth plasticity to light (F = 2.0, df = 270, p<2.2e-16). We quantified growth plasticity as the individual deviation of the genotypic mean of each

## Average regional growth rate

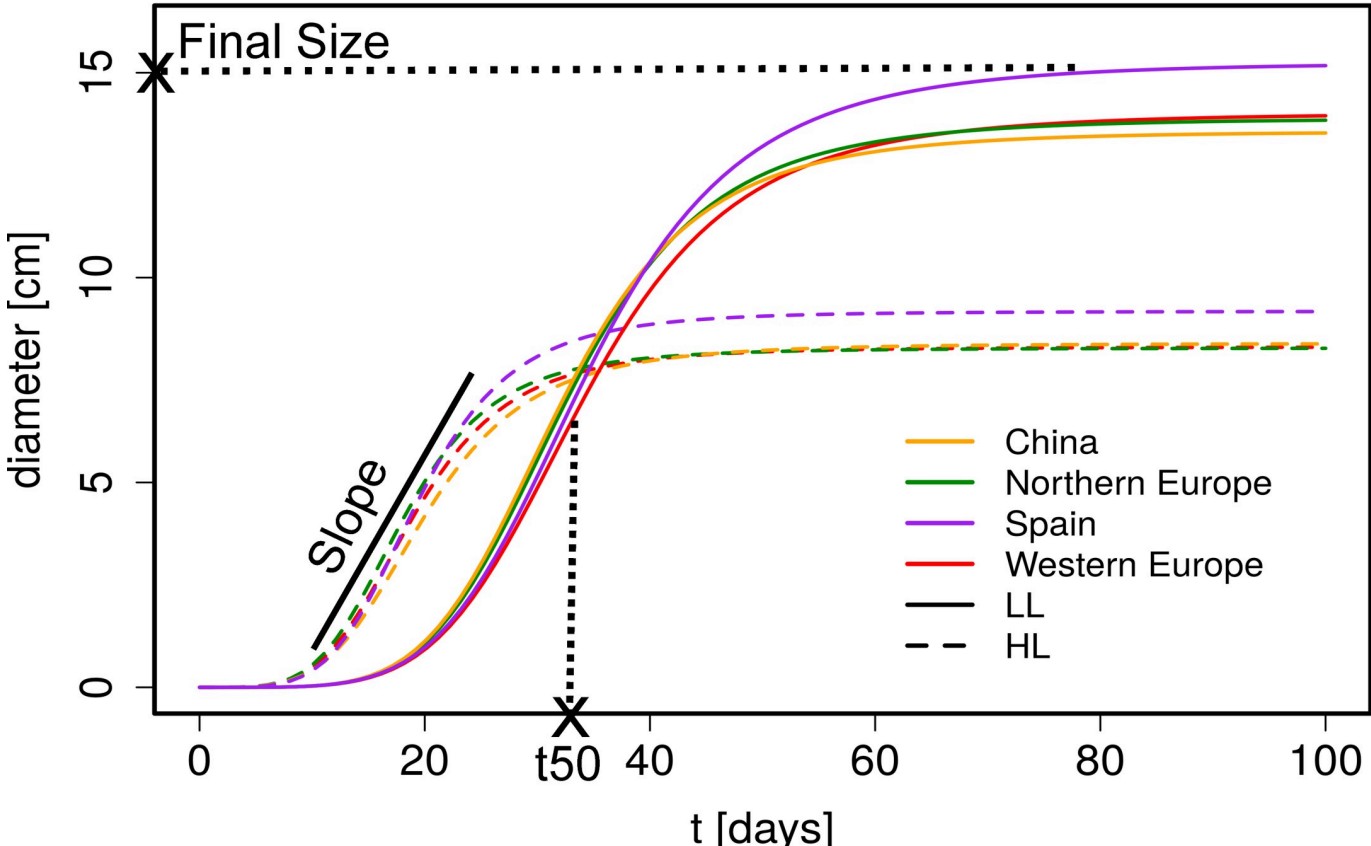

**Fig 1. Regional growth rate estimates in HL and LL.** Predicted growth curves averaged over region (from drm function). The growth curves were estimated from diameter measurements at different time points. Diameter measurements for HL are from day 11 to 46 and for LL from day 24 to 89. An illustration of the parameters that are estimated from these growth curves are included in the plot (Final Size is a diameter, t50 a time point and Slope the fold increase in the linear phase). HL (dashed line), LL (solid line), China (orange), Northern Europe (green), Spain (purple) and Western Europe (red).

genotype in HL and LL from the average reaction of the population to the change in light regime (S8 Fig).

### Spanish genotypes show the most vigorous rosette growth

We found evidence for rosette growth variation across regions (MANOVA in Table 1 and Figs 1 and 2). Within Europe, Spanish genotypes reached the largest FS in both HL and LL plants

**Table 1. Multi- and uni-variate analyses of growth variation in response to light regime, genotype and their interaction.** The multivariate analysis was conducted on the estimates of FS, t50 and slope for all 270 genotypes in three replicates and accounting for block effects nested within light treatment.

| Response | Multivariate analysis (MANOVA) | | | Final Size | | t50 | | Slope | |
|---|---|---|---|---|---|---|---|---|---|
| | df | F | p-value | F | p-value | F | p-value | F | p-value |
| Block | 4 | 27.5 | < 2.2E-16 | 30.15 | < 2.2E-16 | 12.24 | 9.8e-10 | 30.50 | < 2.2E-16 |
| Light regime | 1 | 7388.7 | < 2.2E-16 | 15687.23 | < 2.2E-16 | 9289.44 | < 2.2E-16 | 34.01 | 7.2e-9 |
| Genotype | 279 | 3.6 | < 2.2E-16 | 8.43 | < 2.2E-16 | 2.51 | < 2.2E-16 | 2.036 | < 2.2E-16 |
| Light*Genotype | 270 | 2.0 | < 2.2E-16 | 2.97 | < 2.2E-16 | 1.59 | 2.3e-07 | 1.61 | 4.4e-08 |

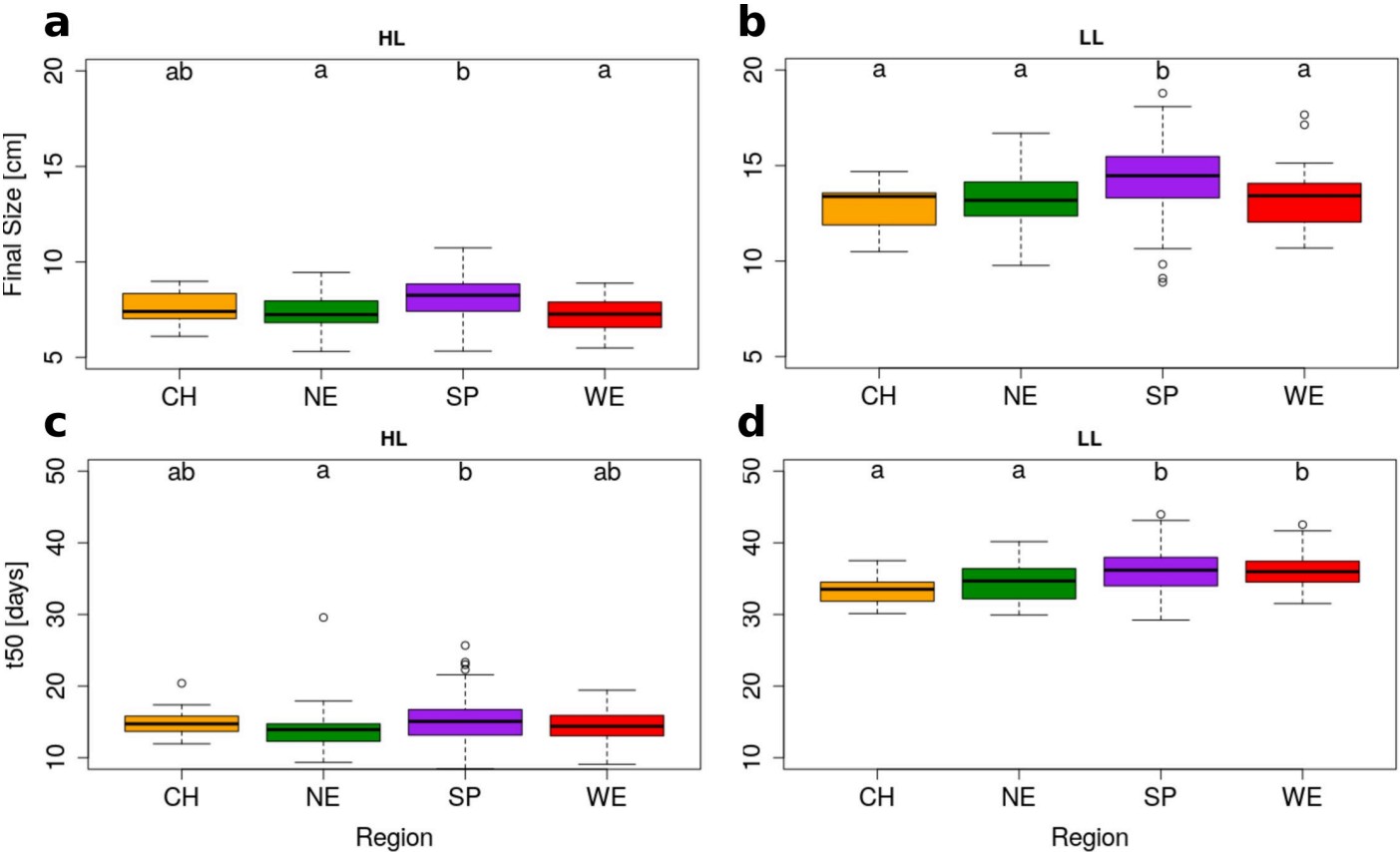

**Fig 2. Significant regional differentiation of Final Size and t50 in HL and LL.** *A.thaliana* genotypes are grouped based on geographical origin. Box plots show regional variation in Final Size (upper row) and t50 (lower row) for HL (left) and LL (right). Groups that do not share a letter are significantly different according to Tukey's HSD (p-value < 0.05). Region information: China (CH, n = 20), Northern Europe (NE, n = 58), Spain (SP, n = 119) & Western Europe (WE, n = 29).

(Tables 1 and S5, MANOVA: F = 16.37, df = 3, p-value = 5.35e-10). Although the growth slope did not differ significantly across regions, we observed that, under HL conditions, Spanish genotypes reached 50% of their FS (= t50) significantly later than the genotypes originating from Northern Europe (t50 = 15.17 vs 13.79, respectively, GLHT z = 3.061, p-value = 0.011, Fig 2C). This effect was also observed for plants grown under LL conditions but we detected no regional difference in GxE (Fig 2D). Since Spain and Northern Europe do not differ in their average flowering time (S9 and S10 Figs), the larger rosette size observed in Spain is not due to an extension of the duration of vegetative growth in this population.

## Chinese genotypes show that growth rate variation is constrained in evolution

Despite a long history of population isolation that was magnified by a strong bottleneck after the last glacial period [38,73], the growth rate of Chinese genotypes was comparable to that shown by most European genotypes (S5 Table and Fig 2A and 2B). Under LL conditions, Chinese genotypes showed lower t50 and FS values only when compared to Spanish genotypes (S5 Table and Fig 2A–2D). Under HL, genotypes from China did not differ significantly from those from any other region (Figs 2 and S11). The analysis of Chinese genotypes indicates that the phenotypic evolution of rosette growth does not scale with the extent of genetic divergence (Fst between Europe and China is 0.057 on average, with a standard deviation of 0.147, and

much greater than Fst between Spain and Northern Europe, KS test, D = 0.39, p<2.2e-16). A parsimonious explanation to the fact that growth rate has not significantly changed despite extensive population divergence, is that the evolution of growth rate is likely to be constrained by stabilizing selection around a growth optimum [1].

The Chinese population was also the only one to show a difference in GxE (S8 Fig). Compared to Spanish genotypes, Chinese genotypes displayed a GxE that was lower for t50 and higher for slope (t50: GLHT z = 2.748, p-value = 0.028; slope: GLHT z = -3.224, p-value = 0.006; S8 Fig). When grown under the LL regime, these genotypes displayed a lower FS than genotypes from Spain. In contrast, within Europe, we observed no significant difference in the growth plasticity of plants in relation to light regime, despite the fact that Northern populations are exposed to lower average light intensity (S3 Fig).

## GWAS reveal only two SNPs significantly associated with rosette growth variation

We used GWAS to determine the genetic basis of variation in growth rate within Europe (Figs 3 and S12–S17). The sample size (15) and strong population structure of Chinese genotypes precluded their inclusion in this analysis (S2 Fig). Henceforth, we focused on the analysis of genetic variation within and among European populations. Overall, we found few significant genetic associations, indicating that genetic variance for growth rate is generally polygenic. One SNP (chromosome 1, position 24783843) associated with t50 variations in LL plants (effect size = -2.475, p-value = 2.6E-9, Fig 3 and S6 Table). A second SNP (chromosome 3, position 951043) was significantly associated with the slope of rosette diameter growth in HL plants within Spain (effect size = 1.229, p-value = 8.4E-7, Fig 3 and S6 Table) and was polymorphic only in the Spanish set of genotypes. This SNP was within a 1Mb DNA fragment showing strong local LD and enclosing 21 genes. Two additional SNPs were associated with GxE for FS and t50 in HL plants in Northern Europe, respectively, with p-values just below the Bonferroni threshold (S6 Table). Yet, we found no SNP significantly associated with FS above the Bonferroni threshold, although FS is the most heritable trait (S4 Table). Diverse genetic setups can result in such polygenic architecture: large effect size variants that are too rare to be detected, many variants with effect sizes too small to be individually significant, or the presence of multiple alleles at causal loci that will blur the genetic association signal [47,74]. Local genetic variation in slope and t50, growth parameters which display moderate but significant genetic variance, appear to be controlled by low-frequency variants of comparatively larger effect, since some of them were associated above Bonferroni threshold (S6 Table). This genetic architecture resembles that reported in the same species for flowering time [75,76]. In contrast to slope and t50, variation in FS appeared more polygenic since it has the highest heritability and no SNP association above Bonferroni confidence levels.

## Polygenic scores and functional enrichments confirm the polygenic basis of growth variation

Traits with polygenic architecture are controlled by variation in many loci of low frequency and/or low effect sizes and dissecting their evolution is arguably a major challenge today in evolutionary biology [77–80]. Specifically, random SNPs with outlier frequency are not always sufficiently corrected for with the kinship matrix and these may give rise to spurious associations. Studies of polygenic traits such as human height have shown that residual effects of population structure can give signals of genetic association [56,81]. Similar effects were also encountered in studies of phenotypic variation in plant systems [57]. They are expected whenever environmental variance co-varies with population structure, as is likely the case in human

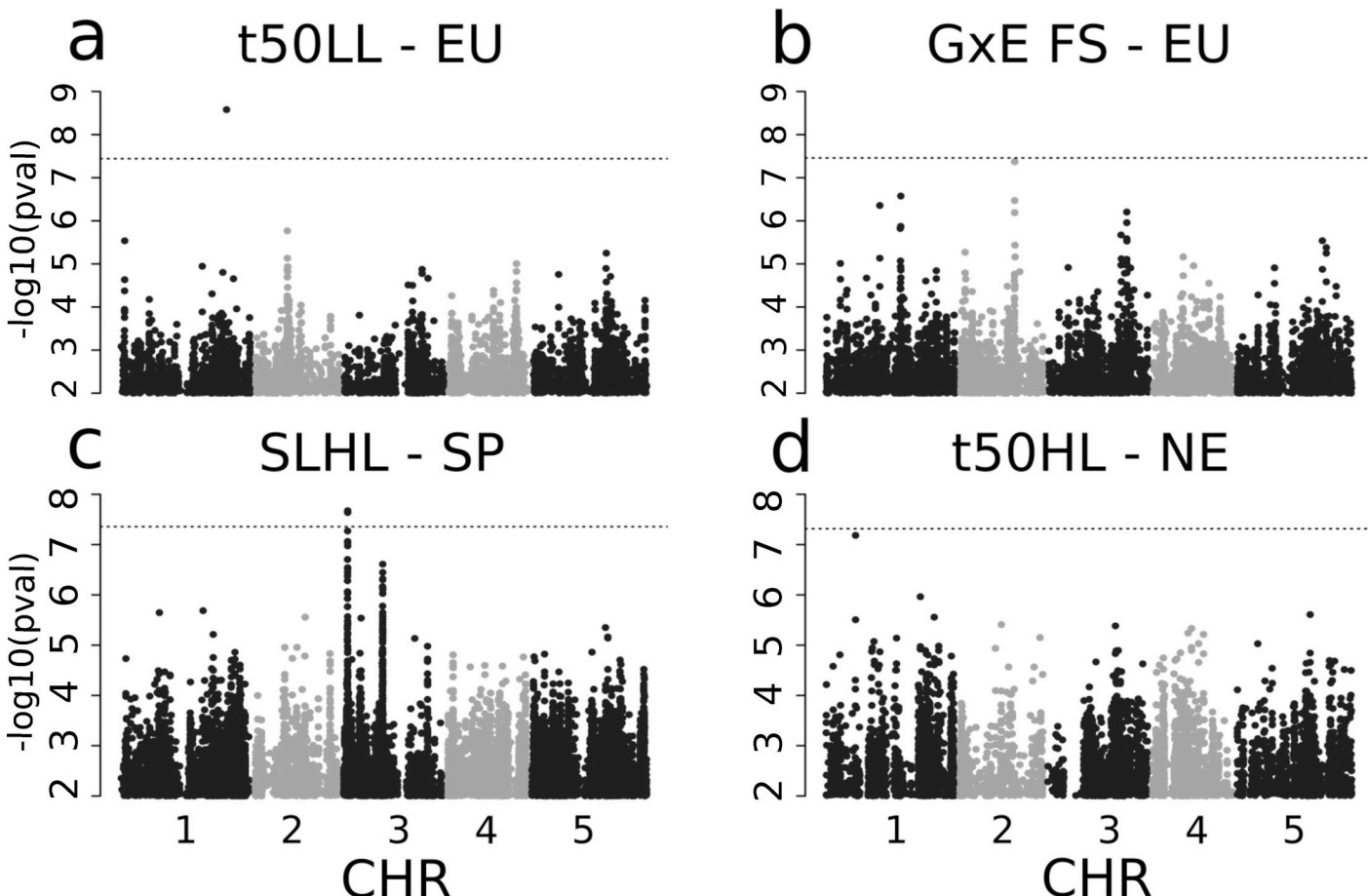

**Fig 3. GWAS-results for 4 phenotypes.** Manhattan plots of GWAS of t50 in LL with all European genotypes (a), with a peak on Chromosome 1, GxE of Final Size with all European genotypes (b) with a peak on Chromosome 2, Slope in HL within Spain (c) with a peak on Chromsome 3 and t50 in HL in Northern Europe (d) with a peak on Chromosome 1. The dotted line shows the corresponding Bonferroni threshold adjusted for a p-value of 0.05.

studies, but can also persist in common garden studies if populations are geographically differentiated in the genetic component of the trait. To confirm the polygenic basis of growth variation, we evaluated the biological relevance of marginally significant genetic associations. The associated sets were composed of 22 to 37 unlinked SNPs. We used their effect sizes to compute polygenic scores for each parameter [53]. We first used to use 80% of the data to identify SNPs associating with rosette growth and test whether they can be used to correctly predict the phenotype of the remaining 20% of the data. This approach, however, did not yield significant predictions (rho = 0.07979094, p = 0.6189), which is not surprising because it usually does not perform well in structured populations [82]. We took a second approach to measure polygenic score accuracy. We used two of the three replicates to compute polygenic scores and tested whether they correlated significantly with the phenotype measured independently in the third replicate (S7A Table). The correlation was highest for FS measured in LL plants (Rho = 0.567, p = <2.2e-16). In fact, FS, the most heritable trait, could be predicted with the highest accuracy in plants grown under both light regimes (S7A Table). When we used random sets of SNPs as input, the computed polygenic scores were significantly correlated with the observed phenotype, indicating that population structure contributes to a significant but small fraction of the variance in polygenic scores. Nevertheless, with this third approach, we showed that polygenic

scores computed on the effect sizes of SNPs associated at sub-significant level were markedly more correlated with the observed phenotype than those computed with random SNP sets (S7B Table). This confirms that sub-significant genetic associations, despite their marginal significance, effectively recapitulate some of the traits' heritability.

We further asked whether sub-significant associations could collectively reveal the specific molecular basis of each trait. We selected SNPs showing a sub-significant association (p<0.0001) and investigated functional enrichment among genes that mapped within 10kb of the SNP. To consolidate our confidence in the functional enrichment, we also pruned tandem duplicates from the annotated set, and determined a p-value threshold that was below the level of significance that can be obtained with GWAS on a permuted data set (see Materials and Methods). While the results reveal many categories without an easily interpretable link to growth, many traits showed functional enrichment within gene ontology (GO) categories, whose link to growth has been documented (S9 Table). For example, genes associated with variation in FS, the most polygenic trait, were enriched among genes involved in the growth-related functions "cotyledon development", "auxin polar transport" and "response to mechanical stimulus"(p-value = 0.0053 or lower). Interestingly, mechanical stimuli have been shown to strongly influence seedling growth, and we observed that FS correlated with hypocotyl length and biomass in 3-week-old plants (S18 Fig, [82]). Additionally, several categories related to defense and stress reactions, such as "response to salt stress", "response to chitin", "regulation of defense response to fungus" and "negative regulation of defense response", were enriched. Variation in stress-related functions is known to have an impact on plant growth in *A. thaliana* [33]. Furthermore, we also found that SNPs associated with FS plasticity to light are enriched among genes involved in the shade avoidance response (*p* = 0.0023), by which plants exposed to limited light conditions increase stem elongation [10,83]. Associated genomic regions included, for example, PHY RAPIDLY REGULATED 2 (PAR2, AT3G58850), a negative regulator of shade avoidance [84] or LONG HYPOCOTYL UNDER SHADE (BBX21, AT1G75540), a regulator of de-etiolation and shade avoidance [85]. Altogether, functional enrichments among genes located in the vicinity of GWAS hits indicated that a biological signal is detectable among sub-significant genetic associations.

As shown above, population structure impacts the results of GWAS and population structure outliers may drive this signal of association. Indeed, genes with elevated Fst reflecting population structure or even regional adaptation of (other) traits could create spurious associations with traits that have a distinct genetic basis but are also differentiated between regions. We thus verified that functional enrichment among genes with SNP associations were different from those observed among genes with elevated Fst. We determined enriched GO categories among genes in the vicinity of GWAS associated loci (p<10−4) and among genes ranked by Fst between Spain and Northern Europe. We visualized overlaps in functional enrichment by clustering GO terms on the basis of the genes they shared (S19 Fig). The enrichment based on Fst revealed three strongly enriched GO terms: "organ morphogenesis", "circadian rhythm" and "virus-induced gene silencing" (p = 0.0009 or lower, S10 Table). The enrichment in GO category "circadian rhythm" may reflect the local adaptation to Northern variations in day length [86,87]. Genes close to SNPs associated with the different growth parameters, however, had clearly distinctive patterns of functional enrichment (S19 Fig and S10 Table). We therefore argue that even though population structure outliers may create some false-positive associations, the polygenic pattern of association that we observe at sub-significant level cannot be explained by the history of population divergence alone.

## No association between per-individual burden and growth

In areas located at the edge of the distribution range of *A. thaliana*, populations may have accumulated an excess of deleterious mutations in the aftermath of their genetic isolation [39,88]. This could have resulted in a mutational load that would have decreased fitness components such as plant growth, because it influences the resources available for the production of progeny [1,20,22]. We thus hypothesized that the lower FS observed in Northern Europe may result from maladaptive forces associated with the demographic history of the region.

This hypothesis could not be supported. No significant difference was detected in total number of LOF mutations per genome in Northern Europe compared to Spain (GLHT: z-value = 0.634, p-value = 0.526, S20 Fig). This observation has been previously reported [55]. In addition, we detected no significant correlation between the number of LOF alleles per genome and the average final size in HL or LL plants within Europe (r in HL = 0.079, p = 0.262, r in LL = 0.029, p = 0.684). Furthermore, we observed no significant difference in growth between Northern European and Chinese populations, despite their significantly higher burden of LOF alleles per genome (GLHT China versus Northern Europe: z-value = -20.259, p-value = <1e-4, S21 Fig, [65]). Therefore, we conclude that the individual burden of LOF mutations is unrelated to rosette growth variation.

We reasoned that lower growth rate might also be associated with a small subset of LOF mutations. To test this hypothesis, we investigated genetic associations between LOF alleles and the three growth parameters (see Materials and Methods). This analysis is similar to a GWAS, but utilises information on approximately 2500 genes that have at least one loss-of-function allele in any of the 1001 Genomes lines [39,55]. We detected no association between LOF alleles and FS, yet there was a significant association of LOF variation at gene AT2G17750 with variation in both t50 in LL plants and t50 plasticity (Fig 4, effect size = -3.542, p-value = 7.49e-6, gene-Fst = 0.113, and effect size = -4.470, p-value = 4.99e-6 for t50 and t50 plasticity, respectively). AT2G17750 encodes the NEP-interacting protein (NIP1) active in chloroplasts, which was reported to mediate intra-plastidial trafficking of an RNA polymerase encoded in the nucleus [89]. NIP1controls the transcription of the *rrn* operon in protoplasts or amyloplasts during seed germination and in chloroplasts during later developmental stages [89]. The LOF variant is present primarily in Northern Europe (MAF = 16 and 0.8% in Northern Europe and Spain, respectively) but is unlikely to be deleterious: it correlates with a decrease of t50, which is a faster entry in the exponential growth phase indicative of increased growth vigor (Fig 4). Taken together, this result does not support the hypothesis that decreased FS in Northern Europe or China is controlled by deleterious variation.

## FS variation might reflect local adaptation at the regional scale

During the growth season, Northern European *A. thaliana* populations are exposed to lower average temperatures (S3 Fig). Smaller rosettes are more compact, and increased compactness is often observed in populations adapted to cold temperatures [32,90–92]. Freezing tolerance, which was indeed reported to be higher in Northern Europe, is associated with functions affecting rosette size [93]. We thus hypothesized that the decreased FS and t50 observed for Northern European genotypes grown under both light regimes is the result of polygenic adaptation to lower average temperatures. We used the 14 to 47 LD-pruned set of SNPs associating in GWAS at a sub-significant level (p<1e-4) to compute polygenic scores for each genotype and each trait, and used Qx, a summary statistic that quantifies their variance across locations of origin. A Qx value outside of neutral expectations inferred from the kinship variance in the population, indicates excess differentiation of polygenic scores, as expected if individual populations evolved under divergent selection [53]. We observed that all traits displayed a strongly

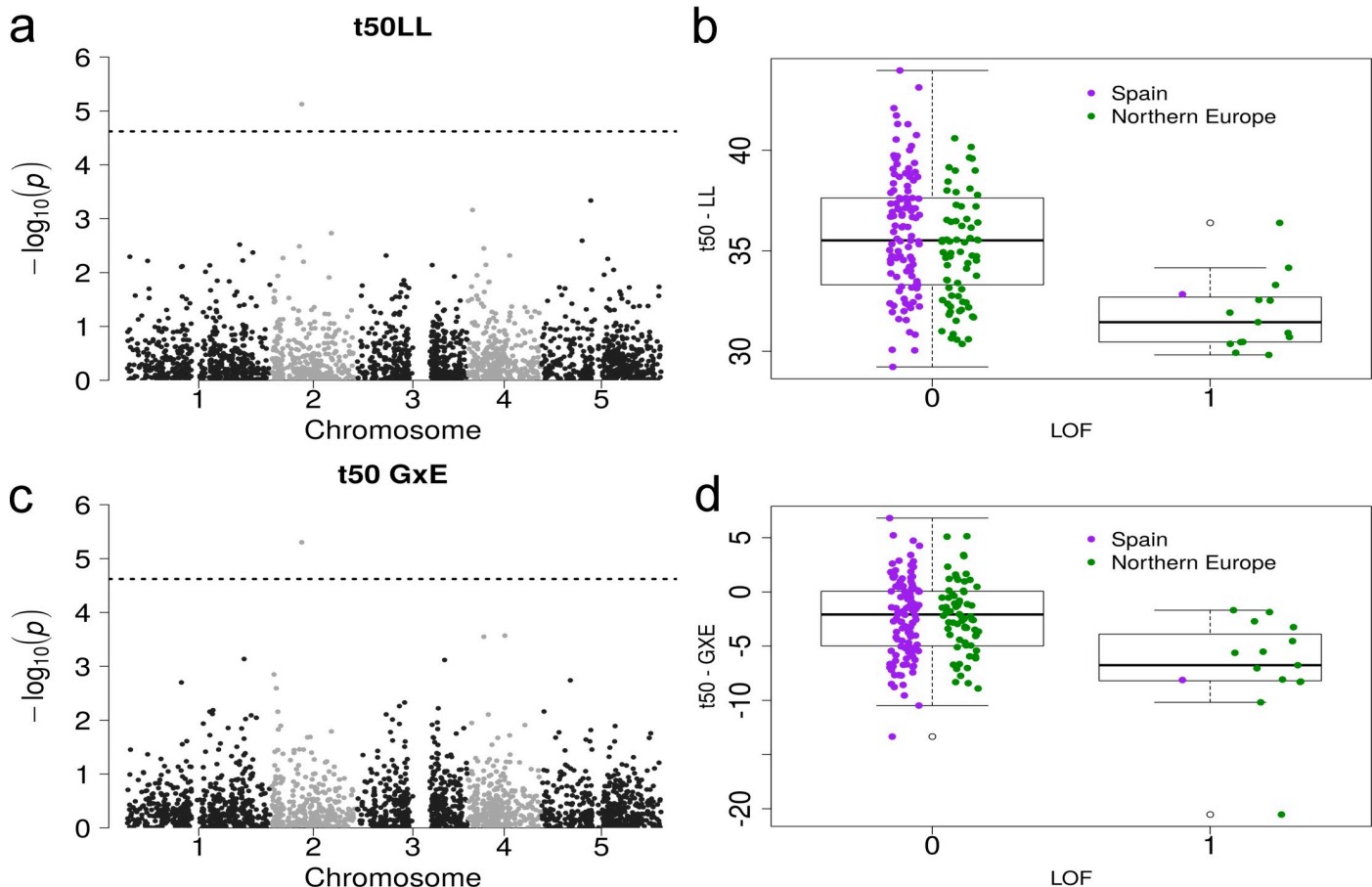

**Fig 4. Loss-of-function association and phenotype of t50 in LL/GxE.** Manhattan plot of a GWAS with loss-of-funcion alleles and t50LL (a) and t50GxE (c) as input phenotypes with the same association (AT2G17750) above the Bonferroni threshold (dashed line). Boxplot of the phenotype of t50LL (b) and t50GxE (d) versus the allele state at AT2G17750 (0 means functional, 1 is a loss-of-function). The colors separate the populations into Spain (purple) and Northern Europe (green).

significant Qx (S8 Table). The differentiation of polygenic scores between the individual populations of origin suggests that divergent selection may be acting locally. Local adaptation has indeed been reported at this scale in this species [94]. This result should however be taken with caution, because, like the GWAS hits it is using, the Qx statistics is sensitive to population structure outliers. Clearly, population structure might underpin more of the GWAS signal detected for slope or t50, which are markedly less heritable than FS.

Interestingly, we observed that FS measured in HL and t50 measured in LL displayed polygenic scores that differed significantly between regions (p-value = 0.0162 and 0.0309, respectively, S22 Fig). We thus further tested whether, at the phenotypic level, regional differentiation in growth rate departed from neutral expectations. We first investigated whether variants associated with phenotypic variation in rosette diameter showed increased genetic differentiation. Compared to the Fst distribution of 10 000 random sets of SNPs, the 95$^{th}$ percentile of 1360 SNPs associating with all three parameters was always higher (p<10$^{-4}$). Thus, associated SNPs are collectively more likely to be differentiated than the rest of the genome. This pattern is not caused by the confounding effect of population structure, because the functional enrichments are mostly specific to the phenotypes (S19 Fig). We note, however, that a few spurious genetic associations could contribute to both higher Fst and over-dispersion of polygenic scores [56–58]. Additional evidence based on approaches independent of

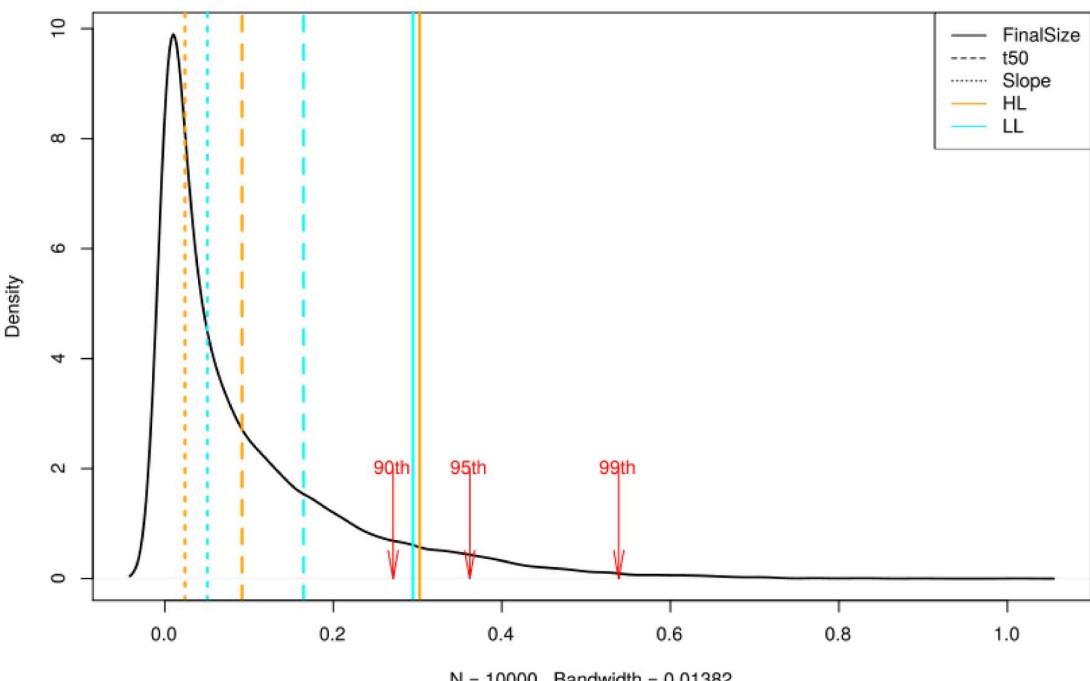

**Fig 5. Expected distribution for quantitative trait differentiation between the Spanish and Northern European population. Qst**. The expectation is based on a multivariate normal distribution assuming a neutral trait with polygenic basis. Vertical lines indicate observed Qst for the individual growth parameters, FS (Solid line), t50 (Dashed line), Slope (Dot line), in HL (orange) and LL (cyan). The red arrows show the 90th, 95th and 99th percentiles of the distribution.

GWAS is therefore required to support the adaptive significance of regional differences in growth rate in Europe. To this end, we used the population kinship matrix to parameterize a multivariate normal distribution and predict the amount of additive phenotypic divergence expected if the trait evolves neutrally [69]. We observed that differentiation for FS measured in HL plants was marginally more differentiated than predicted under neutral conditions (Qst = 0.325, p-value = 0.085, Fig 5). The other parameters did not depart from neutrality (Qst ranging from 0.029 to 0.27, min *p* = 0.11, Fig 5). Since the divergent Chinese population indicates that the unconstrained evolution of growth rate variation is unlikely, this test might be overly conservative. In addition, it predicts the divergence in additive genetic variance, but in the selfing species *A. thaliana*, the whole genetic variance, i.e. broad sense heritability, can contribute to adaptation. We also compared the distribution of phenotypic variation within and between regions to the SNP Fst-distribution [68]. We used the Fst between Northern Europe and Spain as an estimate for nucleotide differentiation and compared it to the differentiation of these populations at the phenotypic level (Qst) [13,45,67]. For FS and t50, the Qst was significantly greater than genetic differentiation at 95% of single nucleotide (Table 2). This suggests that selective forces have contributed to the regional adaptation of FS in Europe. Other climatic components like temperature could also have strong effects on growth differences between populations. Nevertheless, we detected only weak correlations between growth variation and temperature at the location of origin (S23 Fig and S11 Table), suggesting that growth rate could be locally adapted to the conditions prevailing in each region. The environmental factors contributing to adaptive divergence in plant growth thus remain to be determined in this species.

**Table 2. FS and t50 quantitative differentiation (Qst) exceed differentiation given by single SNPs.**

| Trait | Qst | Percentile of Fst |
|---|---|---|
| FSHL | 0.379 | 96.57 |
| FSLL | 0.282 | 95.80 |
| t50HL | 0.300 | 95.95 |
| t50LL | 0.189 | 94.79 |
| SLHL | 0.081 | 93.23 |
| SLLL | 0.010 | 91.62 |

Qst for each trait measured in HL and LL plants. Linear mixed models were used to quantify the ratio of genetic variation between versus within Spain and Northern Europe (Qst). The 95th percentile of the distribution for single SNP Fst between these two regions was 0.205. Permutations confirmed that this test is conservative (see Materials and Methods). HL: plants grown under high light regime, LL: plants grown under low Light regime, FS: Final Size, t50: time to maximum growth and SL: slope.

## Conclusion

Our comprehensive analysis of genetic diversity in rosette growth rate, within and between three broad regions of the distribution area of *A. thaliana*, reveals the environmental and evolutionary factors that control this complex trait, which is of central importance for plant ecology. We show that plastic reactions to light intensity have the strongest impact on variation in rosette growth rates. Yet, we also provide evidence for significant genetic variation within and between regions. We observed that Spanish genotypes show more vigorous rosette growth and reach the larger size, regardless of light conditions. Although GWAS reveal very few associations that pass Bonferroni correction, analyses of functional enrichments and polygenic scores demonstrate that the polygenic basis of trait variation can also be explored in the presence of moderately significant genetic associations. The greater phenotypic differentiation observed within Europe compared to between Europe and China, a pattern opposite to measures of genetic divergence, provides a strong indication that stabilizing selective forces constrain the evolution of growth rate over time. The analysis of polygenic scores and patterns of differentiation suggests that much of the variation observed within Europe has been shaped by natural selection, rather than by the burden imposed by deleterious mutations. Leveraging polygenic associations in local adaptation studies remains challenging [78]. Methodological developments that improve the use polygenic associations for the study of local adaptation are needed to consolidate these conclusions. Understanding the potential of polygenic trait architectures will help better integrate complex traits in our understanding of the genetic processes underpinning ecological specialization [6,95].

## Supporting information

**S1 Fig. Genotype origin map.** Each dot represents the sampling point of a genotype. The genotypes where assigned to Northern Europe (green), Western Europe (red), Spain (purple) and China (orange).
(RAR)

**S2 Fig. Principal component analysis of 227 genotypes.** The PCA is based on 1.5 millions SNPs with a minor allele frequency larger than 0.05. The first two principle components explain about 16% of the variance between the genotypes. Regions: China (orange), Northern Europe (green), Spain (purple), Western Europe (red).
(TIFF)

**S3 Fig. Climatic variation between regions.** A) Annual average of the monthly radiation (left), monthly average temperature (center) and monthly precipitation (right) estimates for the sampling location of each genotype from Worldclim2 data (estimate per ~1km$^2$). Boxplots with different letters are significantly different according to Tukey's HSD (p-value < 0.05). Region information: China (CH, 20 unique locations), Northern Europe (NE, 46 unique locations), Spain (SP, 120 unique locations) & Western Europe (WE, 15 unique locations). B) Experimental Set-up in the growth chamber with the light-spectrum and intensity in HL (left) and LL (right). The bottom bar represent the timing of the light.
(TIF)

**S4 Fig. Projected growth rates and diameter measurements of individual genotypes in HL and LL.** Predicted growth curves averaged per genotype (from drm function). To represent the regions 5 genotypes per region were chosen randomly. The growth curves were estimated from diameter measurements at different time points (points for three input replicates). Diameter measurements for HL are from day 11 to 46 and for LL from day 24 to 89. Legend: Title: Region and ID; HL (dashed line, red points), LL (solid line, blue points), China (orange), Northern Europe (green), Spain (purple) and Western Europe (red).
(TIF)

**S5 Fig. GxE for Final Size.** GxE was estimated based on a glm(Final Size ~ genotype $^*$ environment) and is indicated by the color. Each dot corresponds to a genotype with its phenotype in HL (x-axis) and LL (y-axis) (269 genotypes in total). The black dot shows the average over all genotypes with standard deviation. The line shows a linear model for Final Size in LL ~ Finals Size HL.
(TIF)

**S6 Fig. Regional comparison of the GWAS results within Spain to across Europe.** qq-plots comparing the GWAS data in S13 Fig (Spain, 117 genotypes, x-axis) to the data from S12 Fig (Europe, 201 genotypes, y-axis). The traits are Final Size (upper row), t50 (2nd row) and Slope (lower row) in HL (left column), LL (middle column) and their GxE (right column). The grey dotted line indicates the neutral expectation.
(TIF)

**S7 Fig. Regional comparison of the GWAS results within Northern Europe to across Europe.** qq-plots comparing the GWAS data in S14 Fig (Northern Europe, 83 genotypes, x-axis) to the data from S12 Fig (Europe, 201 genotypes, y-axis). The traits are Final Size (upper row), t50 (2nd row) and Slope (lower row) in HL (left column), LL (middle column) and their GxE (right column). The grey dotted line indicates the neutral expectation.
(TIF)

**S8 Fig. Regional differences for GxE for each trait.** GxE for FS (left), t50 (center) and Slope (SL, right). The phenotypic values are based on 217 genotypes of *Arabidopsis thaliana*. Groups that do not share a letter are significantly different according to Tukey's HSD (p-value < 0.05). Region information: China (CH, n = 14), Northern Europe (NE, n = 58), Spain (SP, n = 117) & Western Europe (WE, n = 28).
(TIF)

**S9 Fig. Flowering time from 1001 Genomes.** Flowering time in 16˚C conditions of each genotype plotted for Northern Europe (green, n =) and Spain (purple, n =, based on data from 1001Genomes, 2016). The regions showed no phenotypic difference, as indicated by the same letter (pairwise GLHT, p-value> 0.05).
(TIF)

**S10 Fig. Flowering time in the experiment.** Flowering time of each genotype in HL (left) and LL conditions(right). Missing values were replaced with 59 (HL) or 90 (LL) days after sowing. Boxplots with different letters are significantly different according to Tukey's HSD (p-value < 0.05). Population information: China (CH, n = 22), Northern Europe (NE, n = 84), Spain (SP, n = 121) & Western Europe (WE, n = 53).
(TIF)

**S11 Fig. Regional differences in Slope.** The phenotypic values are based on 220 genotypes of *Arabidopsis thaliana* in HL (left) and LL conditions (right). Groups that do not share a letter are significantly different according to Tukey's HSD (p-value < 0.05). Region information: China (CH, n = 15), Northern Europe (NE, n = 58), Spain (SP, n = 119) & Western Europe (WE, n = 28).
(TIF)

**S12 Fig. GWAS results for all phenotypes across Europe.** Manhattan plots using 201 (or more) genotypes from Europe (Spain and Northern Europe) as input. The traits are Final Size (upper row), t50 (2nd row) and Slope (lower row) in HL (left column), LL (middle column) and their GxE (right column). The dotted line denotes the 5% Bonferroni-corrected threshold.
(TIF)

**S13 Fig. GWAS results for all phenotypes within Spain.** Manhattan plots using 117 (or more) genotypes from Spain as input. The traits are Final Size (upper row), t50 (2nd row) and Slope (lower row) in HL (left column), LL (middle column) and their GxE (right column). The dotted line denotes the 5% Bonferroni-corrected threshold.
(TIF)

**S14 Fig. GWAS results for all phenotypes within Northern Europe.** Manhattan plots using 83 (or more) genotypes from Northern Europe as input. The traits are Final Size (upper row), t50 (2nd row) and Slope (lower row) in HL (left column), LL (middle column) and their GxE (right column). The dotted line denotes the 5% Bonferroni-corrected threshold.
(TIF)

**S15 Fig. QQ-plots for GWAS results for all phenotypes across Europe.** QQ-plots of GWAS using 201 genotypes from Europe (Spain and Northern Europe) as input. The traits are Final Size (upper row), t50 (2nd row) and Slope (lower row) in HL (left column), LL (middle column) and their GxE (right column). The grey line denotes the neutral expectation and the red line the observation from the data. The axes describe the expected (x) and observed (y) values for -log10(p).
(TIF)

**S16 Fig. QQ-plots for GWAS results for all phenotypes within Spain.** QQ-plots of GWAS using 117 genotypes from Spain as input. The traits are Final Size (upper row), t50 (2nd row) and Slope (lower row) in HL (left column), LL (middle column) and their GxE (right column). The grey line denotes the neutral expectation and the red line the observation from the data. The axes describe the expected (x) and observed (y) values for -log10(p).
(TIF)

**S17 Fig. QQ-plots for GWAS results for all phenotypes within Northern Europe.** QQ-plots of GWAS using 83 genotypes from Northern Europe as input. The traits are Final Size (upper row), t50 (2nd row) and Slope (lower row) in HL (left column), LL (middle column) and their GxE (right column). The grey line denotes the neutral expectation and the red line the

observation from the data. The axes describe the expected (x) and observed (y) values for
-log10(p).
(TIF)

**S18 Fig. Correlation of phenotypic traits.** Pearson correlations for each pair of traits. Colored
boxes show significant correlations (p<0.05 after multiple testing correction (FDR correction)
and correction for populations structure (lmekin)) for 193 genotypes across experiments. The
significance is illustrated by box size (larger box represents lower p-values) and the color
shows the direction and strength of correlation. Abbreviations are: HL = high light,
GxE = Genome x Environment interaction, LL = low light, SL = Slope, FT = Flowering time,
FS = Final Size, DiamFieldM2 = Diameter in Field conditions after 2 Months,
Biomass21d = Biomass in controlled (HL) conditions after 21 days.
(TIF)

**S19 Fig. Functional enrichment dendrogram for GO enrichment.** The enrichment is either
based on ranking genes by p-value of the nearest SNP in GWAS (columns 1–9) or Fst of the
gene (column 10). The GO terms are arranged into 9 clusters of similar function on the right
side of the plot. Depicted are only enrichments with a p-value < 0.001.
(TIF)

**S20 Fig. Loss-of-function alleles per population.** Based on data from Monroe et al. (2018).
The sum of LOF alleles per genotype for Northern Europe (green, n =) and Spain (purple, n
=). The regions were not different from each other (GLHT: z-value = 0.634, p-value = 0.526,
negative binomial distribution).
(TIF)

**S21 Fig. Loss-of-function alleles per population.** Based on data from Xu et al. (2019). Boxplot
of the sum of LOF alleles per genotype for each region. Boxplots with different letters are sig-
nificantly different according to Tukey's HSD (p-value < 0.05). Region information: China
(CH, n = 21), Northern Europe (NE, n = 84) & Spain (SP, n = 121).
(TIF)

**S22 Fig. Polygenic Scores and regional differentiation for each trait.** Summary results from
the analysis after Berg and Coop (2014). Each boxplot depicts the polygenic scores of a trait for
genotypes from Northern Europe (green) & Spain (purple). Boxplots with different letters are
significantly different according to Tukey's HSD (p-value < 0.05). Furthermore, the plot con-
tain information about the number of SNPs used as input, the Qx score for excess variance in
SNPs associated with the trait and the p-value of the Qx-analysis. Traits: FS = Final Size, t50,
SL = Slope, HL = High Light treatment, LL = Low Light treatment.
(TIF)

**S23 Fig. Correlation of phenotypic traits and climate.** Pearson correlations for each pair of
traits/climatic variable. Colored boxes show significant correlations (p<0.05 after multiple
testing correction (FDR correction) and correction for populations structure (lmekin)) for 195
genotypes across experiments. The significance is illustrated by box size (larger box represents
lower p-values) and the color shows the direction and strength of correlation. Abbreviations
are: HL = high light, GxE = Genome x Environment interaction, LL = low light, SL = Slope,
FT = Flowering time, FS = Final Size, DiamFieldM2 = Diameter in Field conditions after 2
Months, Biomass21d = Biomass in controlled (HL) conditions after 21 days, Radiation in kJ/
m$^2$/day, PC1/2_growS = Principle component 1 and 2 of all climatic data in the estimated
growing Season (explaining 88.7 and 10.7% of the variance), PC1/2_T = Principle component
1 and 2 for climatic variables related to Temperature (explaining 98.1 and 1.3% of the

variance), PC1/2_P = Principle component 1 and 2 for climatic variables related to Precipitation (explaining 89.8 and 8.22% of the variance).
(TIF)

**S1 Table. Information on the genotypes used in this study, with their country of origin, assigned region, Genotype name and ID in 1001 Genomes, info on the sampling location and position (latitude and longitude) and the Collector.** In the second part of the table the climatic information on the respective location is summarized with: Number of growing months; in the growing season: average Temperature [˚C], Soil water content [%], Water vapor pressure [kPa], Wind speed [m s-[1]], Radiation [kJ m-[2] day-[1]], Rain [mm]. Afterwards the Bioclim variables 1 t 19 from the Worldclim database (http://worldclim.org/version2). After this the first 2 PCs for PCA on data based on growing season, Temperature variables from bioclim data and precipitation variables from bioclim data.
(XLSX)

**S2 Table. Raw phenotypic measurements for each plant In the experiment.** Replicate is the block the plant was growing in with the corresponding tray number and row and column for position on the tray (5 rows and 7 columns per tray). The "diam" measurements are diameter measurements where the number corresponds to days after sowing.
(XLSX)

**S3 Table. Genotypic mean of each genotype after correction for positional effects.** Information of the usage of genotypes: Phenotype_analysis is 1, if the genotype was used for phenotype-related analysis (regional differentiation, Qst) and GWAS is 1, if the genotype was used in GWAS and following analyses (also GO enrichment & polygenic scores). Additionally data from other experiments that was used for correlations: DiamFieldM2: Diameter in mm in the field in Cologne, after 2 months; Hypocotyllength: length of hypocotyls in mm in HL conditions, 15 days after sowing; Biomass21d: Plant dry weight in g after 21 days after sowing in HL conditions; FT_10/FT_16: flowering time in 10/16˚C from 1001 Genomes, 2016.
(XLSX)

**S4 Table. Estimated heritabilities and pseudo-heritability from EMMAX.** Rows contain the input sample size (N), heritability ($H^2$) and pseudo-heritability for each trait, treatment and population. The p-value of a heritabily is the genotype effect of the mixed linear model.
(XLSX)

**S5 Table. Pairwise comparisons of phenotypes for each trait and treatment.** The mean difference between traits is given with Z- and p-value from a GLHT of a glm(parameter~population).
(XLSX)

**S6 Table. Associated SNPs for the different datasets, traits and environment.** For each associated SNP the Chromosome, Base, minor allele frequency (MAF), -log10(P) and effect size are given. The LD for the focal SNP was estimated, with the number of SNPs and genes within the LD range. The p-value of two SNPs that exceeded the Bonferroni threshold are marked in bold, the others were just below threshold.
(XLSX)

**S7 Table. Testing the accuracy of polygenic trait predictions.** A. Polygenic scores were computed based on the phenotypic measurements for two replicates, and correlated with the phenotype observed for the third replicate. Correlation was tested with a Spearman rank correlation test Rho..Nr_SNPs: number of SNPs associated with each trait at p<10–4. B. SNPs

associated with the phenotype at sub-significant level improve significantly the phenotype prediction but random SNPs show that population structure plays an important role. Rho_associated shows the correlation between polygenic score and the genotypic values. Based on 1000 random samples of an equal number of SNPs, a distribution of random Zscores was computed and compared to the spearman correlation of the prediction of associated variants to the input phenotypes (Rho_associated). The distributions of spearman correlations of the 1000 random sets is described with the median (Rho_random_median), 95$^{th}$ quantile (Rho_random_95-quantile) and the maximal Rho (Rho_random_max). The correlation obtained with random SNP set is also often significant at p<0.05 (Percentage_significant), but the maximum correlation coefficient (Rho_random_max) is always markedly lower than the one obtained with sub-significant SNPs (Rho_associated).
(XLSX)

**S8 Table. Results from Polygenic adaptation test after Berg & Coop (2014).** The trait column contains the respective traits that were used as input and a random set of equal size which was used to predict FSHL in the last row. Qx is the test statistic for a signal of polygenic adaptation using all phenotypic data. Rho are the results from a spearman correlation of Z-scores predicted versus the input phenotypes. The regional Z-values for Northern Europe and Spain are the region specific effect on the trait. P-values from each test are in parentheses. The SNPs column contains the number of input SNPs for the estimation of polygenic adaptation (after pruning).
(XLSX)

**S9 Table. GO-enrichment of genes in LD (within 10kb) to SNPs with p $<$ 0.008 (based on permutation) in a GWAS for the respective trait.** Shown are terms with an enrichment $<$ 0.001. GO.ID and term give information on the enriched GO term. Annotated states all genes that are in the term, Significant is the number of genes that are associated in the input data set and Expected the number of genes that are expected to be enriched by chance. The resultFisher gives the Fisher score for enrichment. We only report GO terms with >5 genes in them.
(XLSX)

**S10 Table. GO-enrichment of all genes ranked by their Fst or p-value of the closest SNP in a GWAS of the respective trait.** Shown are terms with an enrichment $<$ 0.001. GO.ID and term give information on the enriched GO term. Nr_Genes is the number of genes in the respective term. The resultKS gives the Kolmogorov-Smirnov score for enrichment.
(XLSX)

**S11 Table. Loadings of the climate PCAs for S23 Fig.** The input variables for the respective PCA are in the column Climatic_variable and the loading for PC1 and PC2 are in the following columns. The PCAs were performed with data within the projected growing season (PCA_-growing_season, 185 unique locations), for bioclimatic variables related to temperature (PCA_temperature, 180 unique locations) and bioclimatic variables related to precipitation (PCA_precipitation, 180 unique locations).
(XLSX)

**S1 File. R Markdown detailing the statistical analysis of rosette diameter variation.**
(HTML)

## Acknowledgments

We thank Prof. Andreas Beyer and Prof. Arthur Korte for advice regarding GWAS analyses and Emily Wheeler, Boston, for editorial assistance.

## Author Contributions

**Conceptualization:** Benedict Wieters, Sebastián E. Ramos-Onsins, Shamil Sunyaev, Juliette de Meaux.

**Data curation:** Benedict Wieters, Fei He.

**Formal analysis:** Benedict Wieters, Kim A. Steige, Evan M. Koch.

**Funding acquisition:** Juliette de Meaux.

**Methodology:** Fei He, Evan M. Koch, Sebastián E. Ramos-Onsins.

**Resources:** Hongya Gu, Ya-Long Guo.

**Supervision:** Sebastián E. Ramos-Onsins, Shamil Sunyaev, Juliette de Meaux.

**Writing – original draft:** Benedict Wieters, Juliette de Meaux.

**Writing – review & editing:** Benedict Wieters, Evan M. Koch, Sebastián E. Ramos-Onsins, Shamil Sunyaev, Juliette de Meaux.

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
