## [Decision Letter · Decision Letter 0]

3 Jul 2020

Dear Dr de Meaux,

Thank you very much for submitting your Research Article entitled 'Polygenic adaptation of rosette growth variation in Arabidopsis thaliana populations' to PLOS Genetics. Your manuscript was fully evaluated at the editorial level and by independent peer reviewers. The reviewers appreciated the attention to an important problem, but raised some substantial concerns about the current manuscript. Based on the reviews, we will not be able to accept this version of the manuscript, but we would be willing to review again a much-revised version. We cannot, of course, promise publication at that time.

If you decide to revise the manuscript for further consideration at PLOS Genetics, please aim to resubmit within the next 60 days, unless it will take extra time to address the concerns of the reviewers, in which case we would appreciate an expected resubmission date by email to plosgenetics@plos.org.

[LINK]

We are sorry that we cannot be more positive about your manuscript at this stage. Please do not hesitate to contact us if you have any concerns or questions.

Yours sincerely,

Kirsten Bomblies

Section Editor: Evolution

PLOS Genetics

Kirsten Bomblies

Section Editor: Evolution

PLOS Genetics

The reviewers clearly appreciated that this manuscript covers an interesting and important topic, and I agree and would like to see it get to the level that we can accept it. As is, however, there were substantial and I think largely very valid concerns raised. I was similarly puzzled in some sections. Of course this is complex data and a complex study, but I think the reviewers raise good points that will help hone and clarify the story.

Reviewer's Responses to Questions

**Comments to the Authors:**

Reviewer #1: uploaded as attachment

Reviewer #2: The manuscript from Wieters and co-authors sets out to understand the genetic basis of variation for rosette growth in the global population of Arabidopsis. To do so, the authors followed for several weeks the growth, approximated as rosette diameter, of a set of 278 accessions representing four broad geographic regions, grown in two light conditions (HL = high light, and LL = low light). They extracted three parameters from those measurements (FS = final rosette size; SL = slope of the exponential phase of the growth curve; and t50 = time at which rosette diameter is ½ FS) and used them to compare the different regional groups and run GWAS. Finally, the authors used a population genomics approach to look for signals of polygenic adaptation for growth in their dataset (Qx, Qst/Fst).

This manuscript addresses a very important topic in adaptation; the ability to match growth to environmental conditions and available resources is a quintessentially adaptive trait, especially for plants, but still very little is known about how variation for growth is genetically determined. The authors have collected a remarkable amount of phenotypic data, and have conducted an extensive set of analyses. I have, however, a few concerns with some of the analyses or their interpretation, and more in general on how the manuscript is framed. I am listing my main comments below following the organization of the manuscript, although that is not necessarily the order of importance.

- Given the apparent focus of the paper on identifying difference in growth patterns (and local adaptation) between different Arabidopsis populations, I was a bit surprised by the author’s choice of light intensity to explore GXE interactions. While clearly light intensity has a major effect on rosette size, this seems a better setup to study variation in shade avoidance (as the GWAS results confirm) rather than to compare growth habits in different macro-climatic areas. The authors should motivate this choice (over, say, different temperatures, daylengths or water availability), and clarify whether the light intensities used are ecologically relevant (i.e. are consistent with differences between the regions examined in this study).

- The authors use rosette diameter as proxy for growth. I understand of course that that was dictated by technical limitations; total biomass measurements are disruptive, and more comprehensive 3D shape measurements would likely be unfeasible at this scale. However, this means that they are exploring only a particular aspect of growth – depending on their flowering habit, Arabidopsis rosettes will continue to produce new leaves (and therefore “grow”) well past the point where they reach maximum diameter, since new leaves will overlap with older ones but not grow longer than them. Other ways of measuring growth would likely give different results, as it is hinted by the significant, but overall quite limited, correlation between FS and biomass or hypocotyl length. In particular, the effect of light intensity on growth would probably be quite smaller if aerial biomass was measured instead of rosette size; it seems likely that the increase of the latter under low light condition is largely due to the increased petiole length that is known to be associated with shade avoidance syndrome (see for example Sasidharan et al. Plant Phys. 2010).

This does not, of course, invalidate the results presented in the manuscript – however, the authors should make clearer what is the scope of their analyses early in the manuscript, and especially in the abstract.

- The authors use a polygenic score analysis and GO enrichment analysis to show that, while only two SNPs pass the very conservative Bonferroni-corrected significance threshold, the GWAS results describe nonetheless the polygenic basis of variation for the examined growth parameters. I am particularly skeptical of GO enrichment analyses; unless they show clear cut patterns, they are labile to different interpretations. The authors interpretation of the GO enrichment results is that “All traits showed functional enrichment within gene ontology (GO) categories related to growth, confirming that these genetic associations were biologically relevant”. However, many of the enriched categories seem unlikely to be directly associated with variation for growth (e.g. “pollen exine formation”, “nuclear chromosome segregation”, “protein modification by small protein removal”, to name a few). A notable exception is GXE FS, for which the four most significantly enriched categories include “shade avoidance” and other three categories which can be plausibly linked to aerial growth/rosette size. Given how noisy the results are, I would suggest the authors remove this analysis, or limit its discussion to GXE FS.

These considerations are also valid for the comparison between genes with elevated Fst and SNP associations. While this analysis uses a different approach, I would have still expected similar results to the previous analysis for SNP associations; however, there is almost no overlap in enriched GO categories between the two analyses (or, for that matter, between the same parameter measured in different light conditions), which appears to further undermine their value.

As for the polygenic scores, I readily admit to not having extensive experience on the topic, so I might be missing or misunderstanding something – in which case I apologize beforehand. It seems to me, however, that using polygenic scores from two replicates to predict the phenotype of the third replicate is functionally identical to predicting the input data – with the accuracy of the prediction being dependent on how consistent between replicates (i.e. heritable) a trait is. Minimally, the authors should provide evidence that the predictions are significantly different from what you would get for random SNPs (possibly generating a null distribution for each parameter), and clarify the interpretation of the results. While the authors do provide one example using random SNPs for FSHL (Table S11), it is not clear how these SNPs have been selected, and it is not a direct comparison (it appears that the polygenic scores for the random SNPs have been calculated using all three replicates, and not to predict the third replicate as is the case for the results in Table S8).

An even more convincing experiment would be to remove a few accessions from the dataset, perform GWAS and calculate polygenic scores on the rest of the accessions, and then use those polygenic scores to predict the phenotype of the remaining accessions. Given the relatively small sample size, such analysis might however be under-powered.

These observations do not mean that sub-significant association are not biologically relevant; nevertheless, unless the authors can provide a more convincing proof, they should acknowledge the possibility that, especially for the low heritability parameters SL and t50 (0.07-0.02, Table S4), GWAS results are extremely noisy and might not be informative. This, however, could have significant consequences on the interpretation of further analyses that rely on GWAS results (Qx). The authors could consider to only focus on the more heritable FS parameters for most of the analyses.

Other points:

- The authors should give more information on how the accessions were chosen, and how the different European regions were defined - was it based on previous genetic characterization (although West European accessions seem to cluster with either Spain or North European ones in the PCA in Fig. S3), or on climatic differences? The Western European group of accession in particular seems not clearly defined; in Fig. S1, one German accession is classified as West European, but all German accessions are classified as North European in Table S3. At different points of the manuscript is not always clear which regional groups are included in which analyses. Since Western European and Chinese accessions are not included in most analyses after the GWAS, it might be helpful to specify at that point that the remaining analyses will focus on the Spain-Northern Europe comparison, and specifically mention the other groups whenever they are included.

- Data for biomass, hypocotyl length and FS in field experiments are used in comparisons with the rosette growth parameters used throughout the manuscript, but they could be integrated further in the analyses (GWAS), since they would provide additional information on growth variation. Either way, since they are used in the manuscript, data for those experiments should be added to the supplementary material (I don’t think they currently are).

- I am not sure that using the flowering time data from the 2016 1001 Genome Consortium paper to test correlation between flowering time and the growth parameters described in this paper is appropriate here – those plants were not only grown at different temperatures, but also with a different light cycle (16 h of light and 8 of dark vs the 12 h of light, 20ºC, and 12 h of dark, 18 ºC, used for this manuscript). While the general trends would be similar, as the relatively strong correlation between the different sets of flowering time data shows (Figure S19), the differences could be large enough to confound the analysis. While FT16 and FT10 are the next best thing, since flowering time was not measured for all accessions in this study, these limitations should be acknowledged.

- Related to the previous point, the description and presentation of Figure 3 should be improved. Are the thickness and length of the lines proportional to the significance and/or strength of the correlation? If so, the information should be included in the figure (even if, as the figure legend mentions, numeric values are reported in Table S6). The striping is also difficult to see on the red lines.

It also not clear why the “VER” parameter is included, since it is not mentioned elsewhere and it does not seem particularly germane to the analyses in the manuscript. If it is kept in the figure, it should be addressed in the manuscript, and a reference supporting the correlation between FT16-FT10 and vernalization requirements should be added (I could not find mention of that in the 2016 1001 Genomes Consortium paper, but it is a very expansive paper).

- While I have little doubt that growth rate is constrained in Arabidopsis, I am not sure selection is necessarily the only explanation for the observation that growth rates in Chinese populations are within the range of variation within European populations – there could be physiological constraints, or European populations, not having experienced (as a whole) the same bottleneck as Chinese population, could be genetically more diverse (which I believe is the case) and cover a broader range of the phenotypic space for growth. The authors should provide a more detailed explanation of why they think selection plays a major role in this pattern.

- It would be helpful to compare the GWAS results for GXE FS to those for the shade avoidance GWAS (measured as hypocotyl elongation) from Filiault and Maloof, PLoS Genetics 2012.

- I found the LOF GWAS to be an interesting approach, especially since it greatly simplifies candidate gene validation – one could quite safely assume that the LOF, and not some more complex regulatory change, is causal for the phenotypic differences. While I do not expect the authors to start a whole experiment just to humor me, it would be neat and relatively straightforward to knock out NIP1 in a few accessions carrying a functional copy (or, even easier, getting a T-DNA mutant line, since I am guessing the Col-0 allele is functional) and check if t50 is indeed affected.

Minor points:

- Line 49-52: This sentence is a bit confusing, the same concept is expressed more clearly elsewhere in the manuscript.

- Line 224-226: numbers sum up to 231, not 235.

- Line 386-397: Figure S7 only shows that flowering time (as FT16 from the 2016 1001 Genome Consortium paper) is not significantly different between North European and Spanish accessions. Directly showing correlations between FT16/FT10 and the parameters analyzed in the manuscript would be more informative (this regardless of my previous comments on whether using those data is appropriate).

- Line 460: should be 22-37 (in Table S8, there are 37 SNPs in the analysis for t50LL).

- Line 537: should be NIP1, not INIP1.

- Line 553: should be 14-47 (in Table S11, there are 14 SNPs in the analysis for GXEt50).

- Line 563-565: this sentence is likely mis-placed (it is repeated almost verbatim immediately below).

- Line 611-612: While this sentence is technically correct (three Western European accessions were include in the GWAS, and Chinese accessions were included in the LOF analyses), most (population) genetic analyses focus almost exclusively on the Spain/Northern Europe comparison.

- Line 614-615: this should probably be re-phrased to specify that light intensity has the strongest effect among the factor tested in this study (other environmental factors not tested here might have even stronger effects).

- Lines 626-628 and lines 629-630: the two sentences are almost identical.

- Figure 6. “Horizontal lines” should be “vertical lines”.

Reviewer #3: attached

**Have all data underlying the figures and results presented in the manuscript been provided?**

Reviewer #1: Yes

Reviewer #2: **No: **Phenotypic data for biomass and hypocotyl length, and for rosette growth in field experiments are used to validate the main dataset used in the manuscript (lines 356-358), but are not reported.

Information on which accessions carry a functional or LOF NIP1 allele is missing.

The numeric values for Qst shown in Figure 6 are not reported (although I am not sure that is necessary).

Reviewer #3: **No: **raw data (photographs) were not provided

PLOS authors have the option to publish the peer review history of their article (what does this mean?). If published, this will include your full peer review and any attached files.

Reviewer #1: No

Reviewer #2: No

Reviewer #3: **Yes: **Hugo Tavares

---

## [Decision Letter · Decision Letter 1]

12 Oct 2020

Dear Dr de Meaux,

Thank you very much for submitting your Research Article entitled 'Polygenic adaptation of rosette growth in Arabidopsis thaliana' to PLOS Genetics. Your manuscript was fully evaluated at the editorial level and by independent peer reviewers. The reviewers appreciated the attention to an important topic but identified some aspects of the manuscript that should be improved.

We therefore ask you to modify the manuscript according to the review recommendations before we can consider your manuscript for acceptance. Your revisions should address the specific points made by each reviewer.

[LINK]

Yours sincerely,

Magnus Nordborg

Guest Editor

PLOS Genetics

Kirsten Bomblies

Section Editor: Evolution

PLOS Genetics

Thanks for you patience with this very long review process, which was mostly a Covid-causalty. For what it's worth, the manuscript has clearly improved greatly as result of revision!

As you will see from the reviews below, one reviewer is still unhappy with the manuscript, and does not think you have addressed his/her concerns. The points raised are valid, but the discussion is getting rather philosophical, and I think it would be wrong to hold up the paper over something like this. The paper is clearly written, and readers can judge for themselves. However, as you address the other minor comments raised below, please consider changing what you say slightly to accommodate the fact that there are (non-crazy) people who are not convinced by your interpretation of polygenic scores and GO enrichment.

Reviewer's Responses to Questions

**Comments to the Authors:**

Reviewer #1: The revised manuscript by Wieters et al, contains considerable changes which present the conducted research in a clear and understandable way. I also want to thank the authors for clearly answering each of the reviewers' comments.

I agree with all the answers to the comments and with the changes that were made to the manuscript.

Congratulations to the authors with this very nice work.

Reviewer #2: The authors did a great job of streamlining and clarifying the manuscript; both the analyses and the narrative of the manuscript are much easier to follow in this revised version, and this clarified some of the points I found confusing in the original manuscript. Polygenic traits are messy and I really appreciate how thorough the authors are in their approach to this study.

However, some of the concerns I raised in the first round of reviews still stand. As a consequence, I am not convinced that the data, as they are presented, fully supports all of the main conclusions put forward by the authors. Below I am reiterating the points that I think were not fully addressed by the authors.

- In my original comments I had discussed the issues I had with the way polygenic scores results were validated and the interpretation of the GO analyses. I appreciate that the authors have been more cautious in reporting the results of the GO analyses. However, the analyses are largely unchanged, and so is their interpretation (the corresponding section of the manuscript is still titled “Polygenic scores and functional enrichments confirm the polygenic basis of growth variation”, as is, by and large, the way they are reported in the abstract).

As it stands, the polygenic scores results look promising but the way they are being validated (by using two replicates to predict phenotypic values for the third) does not look convincing to me. Both I and reviewer 3 suggested that comparing those results (Table S7) to a distribution based on random SNP sets could solve the problem. As mentioned in my original comments, the one polygenic score calculation for a set of 27 random SNPs presented in Table S8 does not seem very convincing (and I still do not think it can be fully compared to the results in Table S7). Unfortunately, but as the authors themselves point out perhaps not surprisingly, the attempt to predict phenotypic values for a subset of accessions based on polygenic scores was not successful.

I concur with the authors that the definition of GO categories is limited by the availability of experimental evidence for gene functions. While in my mind this makes interpreting results from GO analyses even harder (if we cannot trust the categories, why look at them in the first place), looking at it from this perspective could be interesting if there was independent evidence that the GWAS signal really describes the polygenic basis of those traits (i.e., “I am sure that those associations are meaningful, let’s see if I can learn something about them by looking at GO categories”). However, in this case GO categories are used to validate the goodness of the GWAS results, which I find less convincing (i.e., “since there are significantly enriched GO categories, and some of them could be linked to the traits we are studying, it means that sub-significant SNPs accurately describe the polygenic basis of those traits”). As stated before, the fact that there is no overlap between enriched GO categories for GWAS associations with p < 10-4 (Table S9) and ranked GWAS associations (Table S10), makes me even less confident about the interpretation of these results.

Neither this analysis, not the polygenic score results, seem to definitely confirm that the sub-significant GWAS associations describe the polygenic basis of growth variation.

- Likewise, I am still unconvinced that stabilizing selection is necessarily the explanation for the lack of phenotypic differentiation between Chinese and European (one of the other main results reported in the abstract), instead of, for example, physiological limitations or convergent adaptation. While there is no harm in proposing stabilizing selection as a possible reason for the observed pattern, this particular explanation is reported several times throughout the manuscript.

- In response to one of my comments, the authors added to the abstract and elsewhere that the light conditions they used mimicked latitudinal differences in light intensity, which is useful in better framing the scope of the study. However, it is difficult to compare the light intensity reported for the growth chambers (in µmol m-2 s-1, probably of PAR) to the total radiation reported in Figure S3 (kJ m-2 day-1, which I guess is total radiation), especially not knowing the exact wavelength distribution in the growth chambers. It would be helpful to express those values in a format that is more comparable, and explain more in details how the HL and LL are comparable to latitudinal differences; light intensity in HL is more than double that in LL, while, according to Figure S3, average radiation in Spain is only ~1.5 that of North Europe.

- As other reviewers noticed as well, the division in four group is not always well justified. While I am convinced about the groupings for Spain, Northern Europe and China (based on the PCA in Fig S2), Western Europe does not seem to form a group of its own in that analysis.

As an aside, I realize there was a typo in my previous comments; in point 5), I meant to write that heritability for SL and t50 ranges between 0.07 and 0.2, not 0.02 – sorry if that generated any confusion.

Reviewer #3: In their revised manuscript, the authors amend and clarify several points raised by myself and the other reviewers. Although there haven't been major changes in the content, structure and conclusions of the manuscript, the many changes done throughout seemed to me to clarify the diverse and rich set of analysis done.

The authors' replies and counter-arguments to some of my comments did increase my appreciation for their approach of looking at the results of this garden experiment seen from an evolutionary lens (rather than a purely quantitative genetics perspective). Apart from a few remaining comments below, I think the manuscript as is provides enough clarity to allow readers to draw their own conclusions from the interesting correlations found between rosette growth variation and patterns of genomic variation seen across different subsets of these accessions.

I thank the authors for substantially clarifying the statistical treatment of their trait data (and providing with analysis code). This has really helped me understand their approach and better interpret their figures. I was sorry that my suggestions for alternative analysis didn't prove fruitful, I'm sorry for the time you might have spent on this without good returns. My main intention was to gain an understanding of the uncertainty associated with the growth curves. The authors provide a new supplementary figure S4 with individual curves, which goes some way at communicating this.

I have a few remaining questions, which the authors may wish to address.

L148: very few points for Western Europe are visible in Fig S2, and the few that are visible seem to occur mixed in with Spanish and Northern European accessions (reviewer #2 also highlighted this). Perhaps adding transparency to the points would help visualise them? Also, would it be possible to add number of accessions in each group to the figure legend? Am I correct in understanding that these groupings are based on the "admixture group" provided by the 1001genomes project: http://1001genomes.org/accessions.html? If so, it may be worth explicitly stating so, as this would have clarified my previous concerns about the discretisation of these data (also mentioned by reviewer #1 and #2). I think this is implicitly mentioned in Table S1 legend, but might be worth having it more visibly here in the methods.

L152/L197: The description of the experiment as following a "nested design" might be confusing as some of the factors are crossed and others nested. If I understood the design correctly, I think it is technically a split-plot design (https://besjournals.onlinelibrary.wiley.com/doi/full/10.1111/j.2041-210x.2012.00251.x). Within each treatment the experiment was blocked three times (blocks nested within treatment) and all accessions occurred in each of the 6 blocks (accessions and blocks are crossed). Accession and light treatment are therefore indirectly crossed (i.e. all accession-treatment combinations were done - this is what allowed looking at GxE interaction).

L169/L428: "For genotypes without a flowering individual by the end of the experiment, a flowering time value of 59 or 90 days was assigned to HL and LL plants" As I mentioned before, this is not a correct treatment for censored data. The average (and Tukey test) as calculated by the authors is problematic as it is biased by the number of non-flowering individuals in each group. For displaying these data in Fig S10 the authors could consider plotting those individuals that did not flower as points with a different shape at the upper edge of the y-axis (to clearly indicate they are censored data) and not include them in the boxplot calculation. The authors could report the percentage of individuals that flowered for each population. As the claim is that there are no substantial differences between NE and SP it would be useful to see that there is no substantial differences in the proportion of individuals that flowered in this experiment.

Line 218 "as the ratio of." didn't finish the sentence.

L218: "The heritability of GxE could not be estimated..." As I mentioned, I believe GxE variance could be estimated by fitting a random slopes model, if the authors wish to do so, something like: parameter ~ treatment + (treatment|accession) + (1|treatment:block)

Otherwise, it may be best to omit this statement from the methods.

L432: I still could not follow the argument in favour of stabilising selection (also raised by reviewer #2). The high Fst is a consequence of the separation of these lineages (with the bottleneck further increasing their relative divergence - as within-population diversity must have dropped in Chinese populations). But why would that be expected to result in a change in rosette growth? What level of phenotypic divergence (in a polygenic trait) would be expected to accompany a change in relative genetic divergence? Are other traits in Chinese accessions significantly different from European accessions such that a lack of difference in rosette growth is surprising? Are the data also compatible with stabilising selection across the whole range of Arabidopsis (not just Chinese populations) or evolution around a trait optimum, e.g. due to physiological constraints as reviewer #2 mentioned? Without data on fitness disadvantage of extreme phenotypes, it seems hard to infer whether stabilising selection is at play in this system.

L496: "The associated sets were composed of 22 to 37 unlinked SNPs". Is there any overlap in the SNPs for each trait?

L576: It might be worth mentioning that a common association for T50-LL and T50-GxE is not surprising given the high correlation between these traits (Fig S18). In fact, they are not independent traits, as T50-GxE is calculated from T50-LL and T50-HL: possibly there's lower variation in T50-HL, and therefore variation in T50-GxE is mostly due to variation in T50-LL.

Fig S22: have the authors corrected these tests for multiple testing?

L669: the authors nicely caution that Fst and polygenic scores might be confounded by structure. I was wondering if there is a relationship between Fst and SNP effects (at all SNPs, not just p < 1e-4)? As the authors discuss (L489), it is challenging to disentangle variation that is confounded by structure from variation that is not (despite the statistical adjustment with kinship matrix). I was wondering if looking at this relationship would help in any way to investigate the extent of this issue in these data?

L714: "(blue line)" might be meant as "dashed line".

L756/L762: check numbers of referenced figures

L771/L829/L835: missing sample sizes

**Have all data underlying the figures and results presented in the manuscript been provided?**

Reviewer #1: None

Reviewer #2: Yes

Reviewer #3: Yes

PLOS authors have the option to publish the peer review history of their article (what does this mean?). If published, this will include your full peer review and any attached files.

Reviewer #1: No

Reviewer #2: No

Reviewer #3: **Yes: **Hugo Tavares

---

## [Editor Report · Decision Letter 2]

10 Dec 2020

Dear Dr de Meaux,

We are pleased to inform you that your manuscript entitled "Polygenic adaptation of rosette growth in Arabidopsis thaliana" has been editorially accepted for publication in PLOS Genetics. Congratulations!

Yours sincerely,

Magnus Nordborg

Guest Editor

PLOS Genetics

Kirsten Bomblies

Section Editor: Evolution

PLOS Genetics

Comments from the reviewers (if applicable):

**Data Deposition**

http://datadryad.org/submit?journalID=pgenetics&manu=PGENETICS-D-20-00468R2

**Press Queries**

---

## [Editor Report · Acceptance letter]

20 Jan 2021

PGENETICS-D-20-00468R2 

Polygenic adaptation of rosette growth in *Arabidopsis thaliana*

Dear Dr de Meaux, 

We are pleased to inform you that your manuscript entitled "Polygenic adaptation of rosette growth in *Arabidopsis thaliana*" has been formally accepted for publication in PLOS Genetics! Your manuscript is now with our production department and you will be notified of the publication date in due course.

With kind regards,

Melanie Wincott

PLOS Genetics

On behalf of:
